# Quantifying impacts of the drought 2018 on European ecosystems in comparison to 2003

Allan Buras[1], Anja Rammig[1], Christian S. Zang[1]

[1]Land Surface Atmosphere Interactions, Technical University of Munich, TUM School of Life Sciences Weihenstephan, Hans-Carl-von-Carlowitz Platz 2, 85354 Freising, Germany.

*Correspondence to*: Allan Buras (allan@buras.eu)

**Abstract.** In recent decades, an increasing persistence of atmospheric circulation patterns has been observed. In the course of the associated long-lasting anticyclonic summer circulations, heat waves and drought spells often coincide, leading to so-called hotter droughts. Previous hotter droughts caused a decrease in agricultural yields and increase in tree mortality, and thus, had a remarkable effect on carbon budgets and negative economic impacts. Consequently, a quantification of ecosystem responses to hotter droughts and a better understanding of the underlying mechanisms is crucial. In this context, the European hotter drought of the year 2018 may be considered a key event. As a first step towards the quantification of its causes and consequences, we here assess anomalies of atmospheric circulation patterns, maximum temperature, and climatic water balance as potential drivers of ecosystem responses which are quantified by remote sensing using the MODIS vegetation indices (VI) normalized difference vegetation index (NDVI) and enhanced vegetation index (EVI). To place the drought of 2018 within a climatological context, we compare its climatic features and remotely sensed ecosystem response with the extreme hot drought of 2003. 2018 was characterized by a climatic dipole, featuring extremely hot and dry weather conditions north of the Alps but comparably cool and moist conditions across large parts of the Mediterranean. Analysing ecosystem response of five dominant land-cover classes, we found significant positive effects of climatic water balance on ecosystem VI response. Negative drought impacts appeared to affect a 1.5 times larger area and to be significantly stronger in July 2018 compared to August 2003, i.e. at the respective peak of drought. Moreover, we found a significantly higher sensitivity of pastures and arable land to climatic water balance compared to forests in both years. The stronger coupling and higher sensitivity of ecosystem response in 2018 we explain by the prevailing climatic dipole: while the generally water-limited ecosystems of the Mediterranean experienced above-average climatic water balance, the less drought-adapted ecosystems of Central and Northern Europe experienced a record hot drought. In conclusion, this study quantifies the drought of 2018 as a yet unprecedented event, outlines hotspots of drought-impacted areas in 2018 which should be given particular attention in follow-up studies, and provides valuable insights into the heterogeneous responses of the dominant European ecosystems to hotter drought.

## 1 Introduction

More frequent and longer-lasting heat waves are expected to occur with global warming (IPCC, 2014). If such heat waves coincide with low precipitation sums, so-called 'global-change type droughts' or 'hotter droughts' emerge (Allen et al., 2015; Breshears et al., 2005). In the course of hotter droughts, positive feedback loops related to a non-linearly amplified soil-water depletion through evapotranspiration (Seneviratne et al., 2010) further aggravate surface-temperature anomalies because of reduced latent cooling (Fischer et al., 2007). To emphasize this interdependence of heat and drought and improve projections of potential high-impact events, hotter droughts were recently classified as compound events (Zscheischler et al., 2018). Accounting for the interdependence of climatic drivers for drought, climate model projections generally indicate an increase in the likelihood of a hotter drought during the 21st century (Zscheischler and Seneviratne, 2017). Given the associated climatic properties, hotter droughts are more likely to occur under abnormally stable anticyclonic atmospheric circulation patterns which were recently shown to be connected with a hemisphere-wide wavenumber 7 circulation pattern (Kornhuber et al., 2019). Abnormally stable anticyclonic atmospheric circulation patterns and associated wavenumber 7 circulation patterns have expressed an increasing frequency over the past decades (Horton et al., 2015; Kornhuber et al., 2019).

Hotter droughts feature a wide range of negative impacts on managed and natural ecosystems, e.g. reduced productivity, as indicated by lower vegetation greenness using remote sensing data (Allen et al., 2015; Choat et al., 2018; Ciais et al., 2005; Orth et al., 2016; Xu et al., 2011). As a consequence, agricultural yields decline remarkably during hotter droughts while drought-induced tree mortality increases, with both effects leading to significant economic losses (Allen et al., 2010; Buras et al., 2018; Cailleret et al., 2017; Choat et al., 2018; Ciais et al., 2005; Matusick et al., 2018). Moreover, since gross primary productivity (GPP) decreases during hotter droughts, the resulting lower net carbon uptake may change ecosystems from carbon sinks into carbon sources (Ciais et al., 2005; Xu et al., 2011). However, the response to drought may vary among different land-cover types, particularly between grasslands and forests (Teuling et al., 2010; Wolf et al., 2013).

On the continental scale, the European heat wave of 2003 was long considered the most extreme compound event in Europe over the last century with various impacts on human health (increased mortality particularly in France), economy (decreased crop yield in agriculture and forestry), and ecosystems (reduced productivity, forest die-back, and an increased frequency of forest fires; Fink et al., 2004; García-Herrera et al., 2010). According to Ciais et al. (2005), GPP of European ecosystems was reduced by 30 percent in summer 2003 – a yet unprecedented reduction in Europe's primary productivity which resulted in an estimated net carbon release of 0.5 PG C yr$^{-1}$. Given the wide-ranging impacts, potential climate change feedback loops, and the increasing frequencies of circulation patterns initiating compound events it is pivotal to better understand and thus more

precisely predict the response of managed and natural ecosystems to hotter droughts (Horton et al., 2015; Pfleiderer and Coumou, 2018; Sippel et al., 2017; Zscheischler and Seneviratne, 2017).

In the context of an increased persistence of circulation patterns, the European drought of 2018 is of particular interest. In April 2018, a high-pressure system established over Central Europe and persisted almost continuously until mid of October, thereby
causing a long-lasting drought spell and record temperatures in central and northern Europe. Despite preliminary reports in public news and the world-wide-web (see list of public news references), the direct impacts resulting from the 2018 drought are still unexplored. Consequently, we here quantify the impacts of the extreme drought of 2018 on European ecosystems in comparison to the extreme drought in the year 2003. Thereby, we 1) provide an estimate of European ecosystems immediate response to the drought 2018 in relation to 2003, 2) identify hotspots of extreme drought and associated ecosystem response,
and 3) aim at an improved mechanistic understanding of the processes driving ecosystem responses to extreme drought events.

## 2 Material and Methods
### 2.1 Data sources and preparation
### 2.1.1 Climate data
To visualize the general circulation patterns in 2003 and 2018 we downloaded gridded reanalysis data representing 500 hPa
geopotential height from the NCEP/NCAR Reanalysis project provided by the NOAA climate prediction center (Kalnay et al., 1996) available at the Earth System Research Laboratory (ESRL, https://www.esrl.noaa.gov/). The downloaded data cover the period 1981-2018 at a daily temporal resolution and a spatial resolution of 2.5°. As a representation of high-pressure persistence, we computed the mean geopotential height for each grid cell integrated over four months.

Considering temperature and precipitation, we downloaded interpolated, gridded, monthly minimum and maximum
temperature means as well as monthly precipitation sums from the Climate Research Unit (CRU TS 4.01) covering the period 1901-2018 (Harris et al., 2014) and a at a spatial resolution of 0.5°. For reasons of consistency when standardizing the data, we constrained CRU data to the same period as for geopotential height, i.e. 1981-2018. These variables were used to compute potential evapotranspiration (PET, as defined by Hargreaves, 1994) and the climatic water balance (CWB = P-PET, Thornthwaite, 1948). As done for geopotential height, we for each grid cell and year integrated Tmax and CWB over a four
85 month period as measures of maximum temperature and water balance. We decided to integrate over four months, since integrated values representative of peak season conditions (July and August) covered the majority of the growing season (i.e. April-July in 2018 and May-August in 2003). These specific, differing periods were chosen, since they each represent the peak of drought for the corresponding year (see also section 2.2). Results based on different integration periods (e.g. 3 months, 5 months) did not differ substantially and generally confirmed results based on the four month period.
Processed climate data were spatially truncated to match the region considered for the MODIS satellite images (see next section) resulting in 2521 climate grid cells representing an area of roughly 4.9 million km² and covering 38 years. To allow

for combination with MODIS data throughout the analyses, processed climate data were re-projected to MODIS native projection using zonal means while retaining the spatial resolution of 0.5°.

### 2.1.2 MODIS vegetation indices

Using the Application for Extracting and Exploring Analysis Ready Samples (AppEEARS; https://lpdaacsvc.cr.usgs.gov/appeears) we downloaded two MODIS vegetation indices (VI, i.e. the Normalized Difference Vegetation Index NDVI and the Enhanced Vegetation Index EVI) and the corresponding pixel reliability layers at 231 m spatial resolution and 16 days temporal resolution in their native projection. The downloaded data span the period from 100 February 2000 until end of 2018 and cover the area between 10° E and 30° W longitude, 36.5° N and 71.5° N latitude, corresponding to the spatial extent of the CORINE land cover information (see section 2.1.3) .

Based on the pixel reliability information, we only retained records with good or marginal quality for subsequent analyses. Consequently, for most of the grid cells the VI time series contained missing values due to temporary clouds or snow cover. If the number of missing values was larger than the number of VI records, we considered the representing records as insufficient 105 for our analyses and consequently removed the corresponding pixel from the analysis. However, since we were only interested in VI time series during the growing season, we only considered the period from beginning of March (DOY 64) to end of October (DOY 304) for the definition of valid pixels. Following these selection criteria, we retained 95,523,236 pixels for the final analyses, representative of an area of 5,097,215 km².

Prior to the analyses, VI time series of the retained pixels were further processed. We linearly interpolated the missing values 110 of the corresponding VI time series for each pixel using the previous and succeeding records (Misra et al., 2016, 2018). To visualize the potential influence of this gap-filling procedure, we provide Fig. S1 which depicts the number of gaps filled in 2003 and 2018. Since more than 66% of the pixels rendered one or zero gaps, and 95 % rendered five or less gaps (i.e. less than one third of corresponding images missing, thus presumably sufficient data for meaningful interpolation) we assume any potential biases caused by the gap-filling procedure to be marginal. Subsequently, we removed negative outlier values from 115 each VI time series by computing standardized residuals to a Gaussian-filtered (filter size of 80 days, i.e. 5 MODIS time steps), smoothed time-series. Residuals exceeding two negative standard deviations were replaced by the equivalent value of the smoothed time series (see also Misra et al., 2018, 2016). We smoothed the interpolated, outlier-corrected time series by reapplying the Gaussian filter. This procedure was necessary to efficiently handle the remaining high-frequency variability in the seasonal VI-cycle (Misra et al., 2016, 2018).

Finally, VI time series were detrended individually for each pixel by determining the linear trend of VI for each pixel and subtracting the pixel-specific trend from the corresponding pixel. This detrending was necessary to compensate trends that were reported for vegetation indices (Bastos et al., 2017) which were also apparent in the downloaded data (Fig. S2). A comparison between non-detrended and detrended data revealed similar spatial patterns with respect to between-pixel variability, however with amplified differences between 2003 and 2018 in the raw, non-detrended data. That is, for the raw 125 data, the observed trend caused - depending on its sign - lower or higher peak-season VI values in 2003 compared to 2018,

thereby introducing an offset between these two drought events. Concluding, the detrending was able to efficiently handle the varying VI-trends over the MODIS-era, while spatial patterns were generally retained.

Both NDVI and EVI are considered as proxy for photosynthetic carbon fixation, and thus allow for assessing possible changes in productivity in dependence of environmental conditions (Huete et al., 2006; Myneni et al., 1995; Xu et al., 2011). NDVI has earlier been used in the context of drought monitoring (Anyamba and Tucker, 2012) and assessing impacts of drought on ecosystems on large scales (Orth et al., 2016; Xu et al., 2011). While NDVI relies on information derived from the red (RED) and near infrared (NIR) spectra (see equation 1) EVI additionally makes use of the blue (BLUE) spectrum (see equation 2) to reduce atmospheric disturbance and influence of the understory:

$$NDVI = \frac{NIR - RED}{NIR + RED} \quad (1)$$

$$EVI = G \cdot \frac{NIR - RED}{NIR + C_1 \cdot RED - C_2 \cdot BLUE + L} \quad (2)$$

With G being the gain factor, $C_1$ and $C_2$ being the spectrum-specific coefficients of the aerosol resistance term, and L the canopy background adjustment term (G = 2.5, $C_1$ = 6, $C_2$ = 7.5, L = 1 for MODIS EVI, see Huete et al., 2002). Given these definitions, NDVI is more chlorophyll sensitive, while EVI is more sensitive to canopy structural variations (Huete et al., 2002). Thus, NDVI is more likely to reflect changes in leaf coloration as for instance in course of premature leaf senescence under drought, whereas EVI may better reflect early leaf shedding. For reasons of simplicity and to render our results comparable to previous studies which all used NDVI, we focus on results derived from NDVI. To provide the full picture, results derived from EVI are shown in the supplementary information, generally confirming the results based on NDVI.

### 2.1.3    CORINE land cover information

To get an impression on the drought-impact on key European ecosystem components, analyses were stratified using the Coordinated Information on the European Environment land cover map (CORINE, https://land.copernicus.eu/pan-european/corine-land-cover) at 100 m resolution. Since land cover may change over time, we used two different time-steps of the Corine land-cover map, i.e. representative of 2000 and 2018. The land cover maps were re-projected (as were the gridded climate data) to MODIS native projection and resolution using the nearest neighbour method, thereby retaining the original land-cover classes. Given their dominance in Europe and their importance for land-use, we constrained this stratification to pastures, arable land, as well as coniferous, mixed, and broadleaved forests. Consequently, all pixels which either changed their land-cover from 2000 to 2018 and/or did not belong to the five selected land cover types were discarded from further analyses. Based on this selection procedure, we eventually retained 46,908,871 pixels, representative of an area of 2,503,104 km². Figure S3 depicts all pixels used for further analyses as well as their specific land cover class.

### 2.2    Statistical analyses

To quantify weather conditions for the years 2003 and 2018 in relation to average conditions, standardized anomalies of 500 hPa geopotential height, Tmax, and CWB were calculated. To account for seasonal-specific climate parameter distributions, standardization was performed month-wise, i.e. for instance for all CWB-January values. Before doing so, we tested the underlying assumption of normal distribution by computing Shapiro-Wilk test for each grid cell and climate parameter respectively (Fig. S4). The number of significant tests ($p < 0.001$) indicating non-normal distribution was in the order of expected false positives (0.0-0.1 percent vs. 0.1 percent type I error probability). Therefore, we considered the assumption of normality to be fulfilled. To derive anomalies, we first computed the mean and standard deviation for all variables for the full period (1981-2018). Subsequently, we determined the difference between 2003 and 2018 of the respective metric to its corresponding mean in units of standard deviations which in the following are called standardized anomalies (this procedure is also known as z-transformation). To assess the spatio-temporal development of Tmax and CWB for the two distinct drought events, we first of all mapped and evaluated integrated Tmax and CWB from January through October for 2003 and 2018 (see supplementary videos V1 and V2, available at http://doi.org/10.5446/44027 and http://doi.org/10.5446/44028). Moreover, we also pooled Tmax and CWB anomalies for Northern Europe (north of 55°N latitude), Southern Europe (south of 45°N latitude), and Central Europe (in between) and compared the temporal development of mean anomalies between 2003 and 2018 for the three regions. Both Tmax and CWB indicated that the drought of 2003 peaked in August, whereas the drought of 2018 peaked in July (Fig. S5). Consequently, to compare ecosystem response during the peak of drought we focused our comparison between 2003 and 2018 on these two time-steps, i.e. August 2003 vs. July 2018.

Thus, for integrated geopotential height, Tmax, and CWB we eventually used each one standardized anomaly per grid cell for August 2003 and July 2018. The resulting standardized anomalies were mapped and statistically evaluated using histograms. Histograms were used to depict the absolute area representing climate anomalies in 2003 and 2018 which were compared among each other as well as to a normal distribution as a representation of conditions as expected in years representative of average conditions.

To quantify the response of European ecosystems to the two drought events, we focused on end-of-August (DOY 241) and end-of-July (DOY 209) VI values for 2003 and 2018, respectively. The selection of these particular dates was based on the peak of climatological drought (see previous paragraph). Since VI features a bounded distribution (values between -1 and +1), we could not apply a standardization approach as for the climate variables. Therefore, we for each VI time series computed its quantiles over the 19 years similar to Orth et al. (2016). The corresponding quantiles were mapped for August 2003 and July 2018. Areas representing the 19 different quantiles were extracted and compared between August 2003 and July 2018 in a histogram. To depict the temporal development of drought responses in 2003 and 2018, corresponding VI quantiles of areas featuring a CWB anomaly lower than -2 were averaged for each time-step over a compromise of growing seasons, (i.e. beginning on May 9th until November 1st with 16 days interval) and visually compared to each other. To complement these

analyses, we also mapped and evaluated VI quantiles from May 9th to November 1st for 2003 and 2018 in a similar manner as for Tmax and CWB (see supplementary videos V3 and V4, available at http://doi.org/10.5446/44029 and http://doi.org/10.5446/44030). These spatiotemporal analyses generally confirmed the selection of August 2003 and July 2018 to represent the peak of drought impact on selected ecosystems.

Since we were aiming at a better understanding of particular ecosystems' response to drought severity, we subsequently pooled VI quantiles according to three classes of CWB anomalies (abnormal water deficit: CWB < -2, weak water deficit: -2 < CWB < 0, no water deficit: CWB >0) and the chosen five CORINE land-cover classes (arable land, pastures, coniferous forest, mixed forest, broadleaved forest). For the resulting 15 combinations, we compared the areas representing the 19 different quantiles between 2003 and 2018 as done for the total scene. Since the areas of CWB-land cover combinations differed between 2003

and 2018, we moreover computed histograms expressing proportional areas for 2003 and 2018. To get an impression on the absolute share of land cover types in regions that were defined as being under drought in August 2003 and July 2018 we provide Figure S6. Due to the different spatial patterns in drought severity between the two years, we further determined the intersection area where the same CWB anomaly classes were observed in both 2003 and 2018. For this intersection area, we repeated the comparison of the 19 different quantiles between 2003 and 2018 for the five different land-cover classes by

comparing the same pixels for each combination of land-cover class and CWB anomaly class between the two years. This was done, to avoid artefacts related to the fact that CWB anomaly classes were represented by different ecosystems in 2003 compared to 2018. Since the overlap for positive CWB anomalies was very low (altogether only 455 MODIS pixels, and thus no observation for some of the quantiles in some of the land cover classes), we refrained from computing corresponding histograms given their low representativity. Consequently, we only depict the comparison for extreme (CWB < -2) and

moderate (-2 < CWB < 0) water deficit. Because of similar areas in both years, we refrained from depicting proportional areas as for the full comparison.

Finally, we aimed at developing empirical relationships between CWB anomalies and VI quantiles for the five CORINE land-cover classes mentioned before. For this, we logit transformed VI-quantiles (quantiles ranging from 0 to 1) to obtain an unbounded distribution and subsequently extracted the corresponding mean of transformed VI-quantiles for each CWB grid-

cell (thus n = 2521). To assess the effect of different land cover classes, we extracted both the mean VI-quantiles representing all five land cover classes as well as for each land cover class separately. For the corresponding 2521 CWB-pixels we computed linear regressions between the transformed VI-quantiles as the dependent variable and CWB anomalies as independent variable separately for 2003 and 2018 and for the six different land cover types (i.e. five separate classes as well as their combination). For linear regression evaluation, we report adjusted $r^2$ and display scatterplots of logit-transformed VI quantiles vs. CWB along

with the corresponding regression line. Moreover, regression slopes were compared statistically for each land cover type between August 2003 and July 2018. For this, each slope estimate was bootstrapped using random subsampling over 1000 iterations and the overlap of 99.9 % confidence intervals was evaluated. That is, in case the confidence intervals of a respective comparison did not overlap, we considered the difference between slopes as significant. In a similar manner, we compared model slopes among ecosystems (i.e. pastures, arable land, as well as coniferous, mixed, and deciduous forest) separately for

August 2003 and July 2018. Model slopes were grouped according to their overlap of 99.9 % confidence intervals. Finally, to backup observations made from our analyses we also computed a global model (n = 5042) by applying a linear mixed effects model (lme) to VI quantiles using climatic water balance as fixed effect and incorporating crossed random slopes of land cover and year. All analyses were performed in 'R' (R core team, 2019) extended for the packages, 'nlme' (Pinheiro et al., 2017), 'raster' (Robert J. Hijmans, 2017), and 'SPEI' (Beguería and Vicente-Serrano, 2013).


## 3 Results

All considered climate parameters indicated abnormal weather conditions for July 2018 (Figs. 1-3). The integrated 500 hPa geopotential height, an indicator of the persistence of the atmospheric circulation, expressed anomalies in the order of two positive standard deviations for large parts of Central and Northern Europe, mainly covering the Baltic Sea region (Fig. 1 b).

In comparison, July 2018 differed from August 2003 by featuring a dipole of 500 hPa geopotential height anomalies. While in August 2003 most of Europe featured positive anomalies, the Mediterranean was characterized by negative geopotential height anomalies in 2018 (Fig. 1 a vs. Fig. 1 b). The observed dipole of July 2018 expressed a bimodal distribution of anomalies, while August 2003 featured a skewed distribution towards positive anomalies (Fig. 1c). Consequently, in July 2018 the area featuring positive anomalies was 61% of the area in August 2003, i.e. 1.8 million km² vs. 3.0 million km². At the same time,

the area with negative anomalies was 2.4 times higher in 2018, i.e. 2.0 million km² in 2018 vs. 0.8 million km² in 2003 (Fig. 1c).

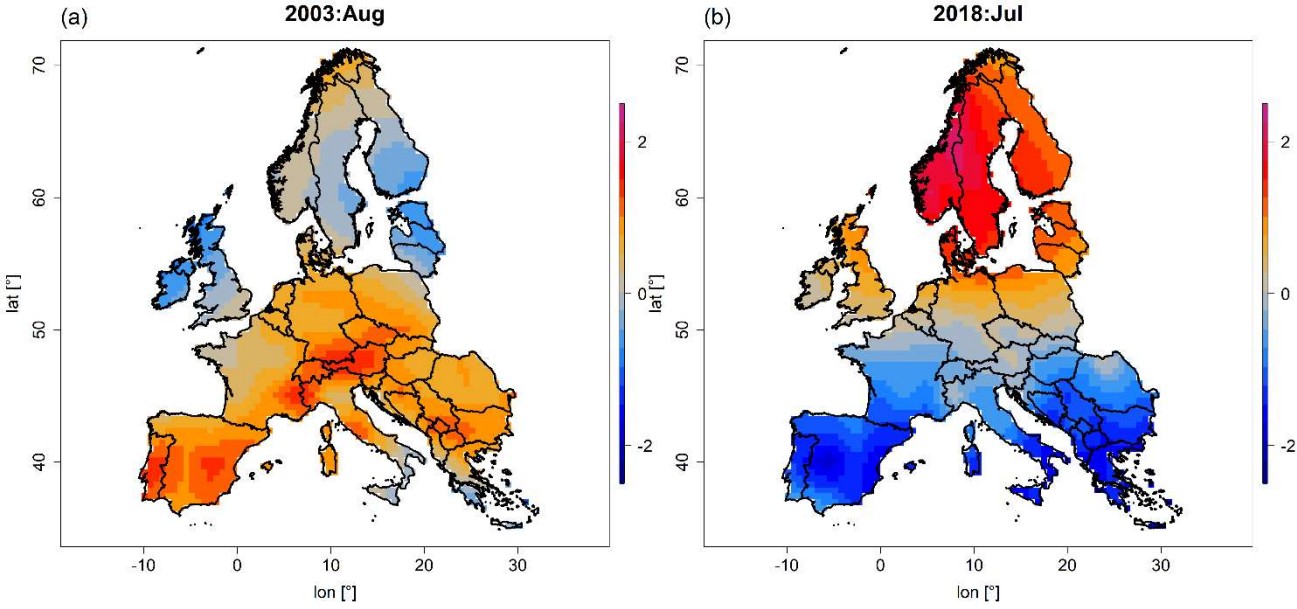

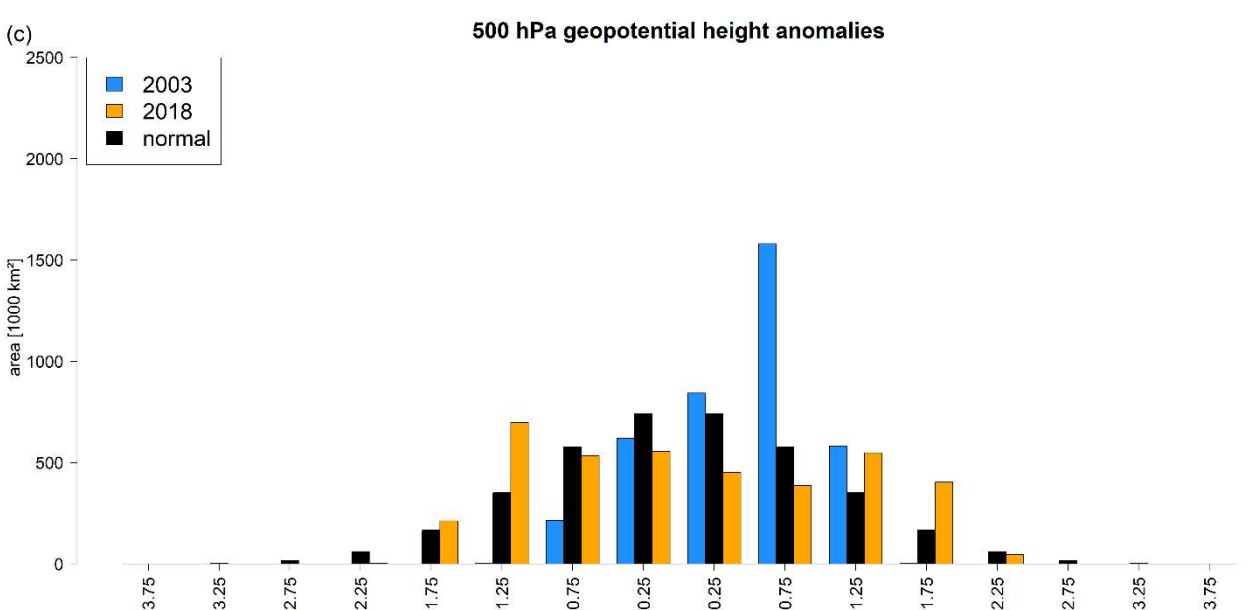

**Figure 1: Maps depicting standardized anomalies of 500 hPa geopotential height for August 2003 and July 2018 (a, b) as well as corresponding area histograms (in units of 1000 km²) for 2003 (blue), 2018 (orange), compared to a normal distribution (black) (c). Blue colours in (a) and (b) indicate geopotential height lower than average, whereas red colours indicate above average geopotential height in comparison to the 1981-2018 mean.**


Anomalies of Tmax revealed up to four positive standard deviations (i.e. extreme heat) over large parts of Central and Northern Europe in 2018 (Fig. 2 b). In contrast, the Mediterranean featured average conditions (i.e. slightly warmer or cooler) and negative anomalies on the Iberian Peninsula. Although the total area with positive Tmax anomalies was more or less similar in August 2003 and July 2018 (Fig. 2c), anomalies above two positive standard deviations covered a 1.9 times larger area in 2018, i.e. 4.0 million km² vs. 2.1 million km² in 2003 (Fig. 2c). Most contrasting differences between August 2003 and July 2018 were observed on the Iberian peninsula (hot in 2003, cool in 2018) as well as Scandinavia and the Baltic Sea region (average in 2003, hot in 2018). The temporal assessment of Tmax anomalies revealed that particularly Northern Europe but also Central Europe experienced strong positive anomalies in 2018 that were higher compared to 2003 (Fig. S5). For Southern Europe the opposite was observed, i.e. less extreme Tmax anomalies in 2018 compared to 2003. Regarding the timing, the peak of the heatwave occurred one month earlier in 2018, i.e. July vs. August in 2003.

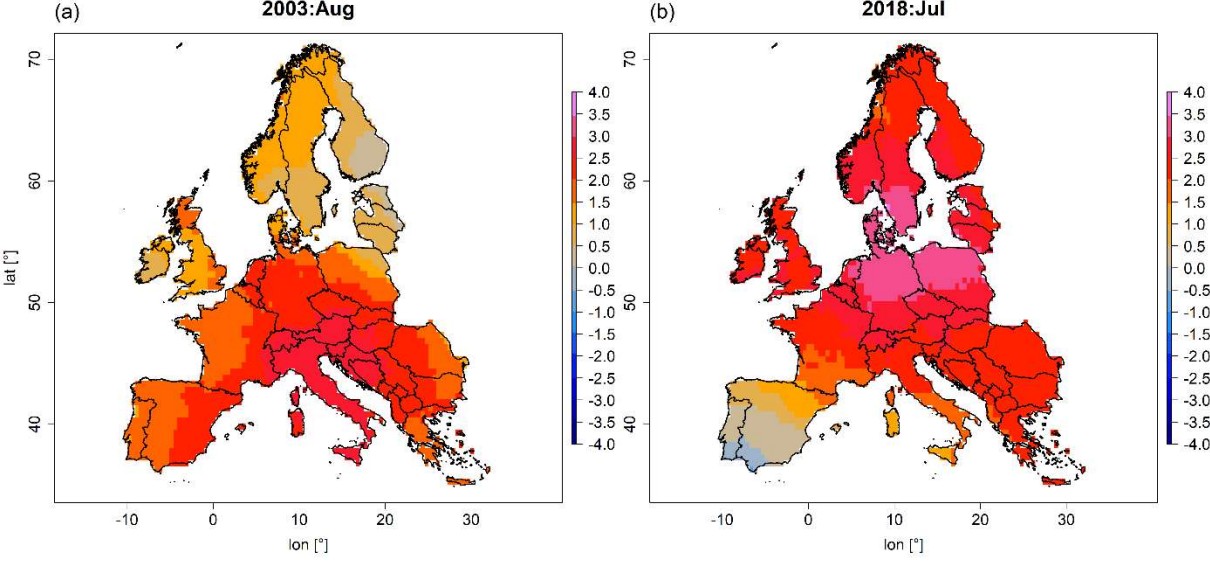

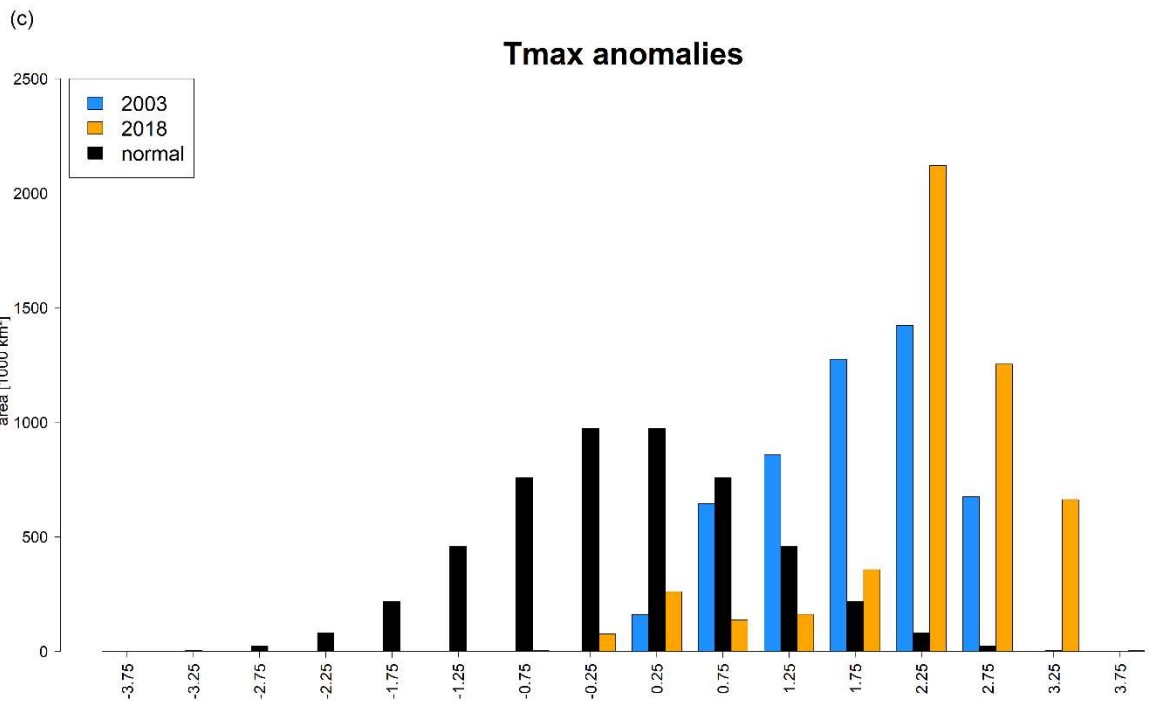

**Figure 2: Maps depicting standardized anomalies of Tmax for August 2003 and July 2018 (a, b) as well as corresponding area histograms (in units of 1000 km²) for 2003 (blue) and 2018 (orange), compared to a normal distribution (black) (c). Blue colours in (a) and (b) indicate relatively cool conditions, whereas red colours indicate warmer conditions in comparison to the 1981-2018 mean. Time-series depicting the temporal development of Tmax anomalies for Northern, Central, and Southern Europe are shown in supplementary Fig. S5. A video depicting the spatio-temporal development of Tmax anomalies from January through October is provided in supplementary V1 (http://doi.org/10.5446/44027).**

For 2018, CWB revealed patterns largely consistent with Tmax (Fig. 3b). Again, Central and Northern Europe featured extreme negative (thus dry) deviations, while the Mediterranean generally expressed positive (thus moist) deviations. In comparison (Fig. 3b vs. 3a), the area with negative (i.e. dry) CWB anomalies was slightly higher in 2003, i.e. 4.2 million km² in August 2003 vs. 3.9 million km² in July 2018 (Fig. 3c). However, if considering CWB anomalies below two negative standard deviations (i.e. extreme drought), in July 2018 an area 1.4 times larger than in August 2003 was affected, i.e. 1.1 million km² vs. 0.8 million km² (Fig. 3c). Consequently, the contribution of the five land cover types under consideration featuring CWB anomalies below two negative standard deviations, revealed a 1.3 times higher overall spatial contribution in July 2018 compared to August 2003, i.e. 610,000 km² vs. 460,000 km² (Fig. S6). These differences were mostly related to coniferous forests due to the more northern location of the drought 2018.

Regarding the spatiotemporal assessment of CWB anomalies, 2003 and 2018 featured a contrasting development (Fig. S5 and supplementary video V2). While Northern Europe experienced strong water deficit from May through August in 2018 and more or less normal conditions in 2003, Southern Europe was characterized by pronounced water deficit from June to September 2003 while it showed positive anomalies throughout the whole growing season in 2018. Central Europe, however expressed more or less comparable conditions at the peak of drought (August 2003 vs. July 2018), but here, drought conditions began earlier in 2003 compared to 2018. Concluding the climatological assessment of the two drought events, 2003 featured a later peak of drought (August) and was centred around Southern and Central Europe while 2018 featured an earlier peak of drought (July) that was centred around Central and Northern Europe.

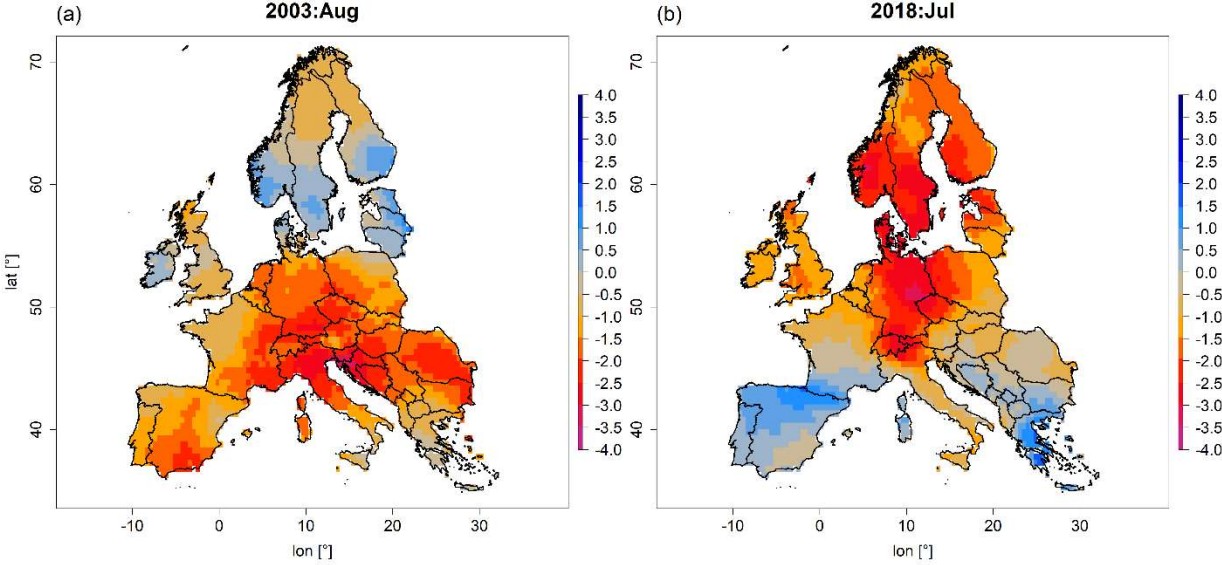

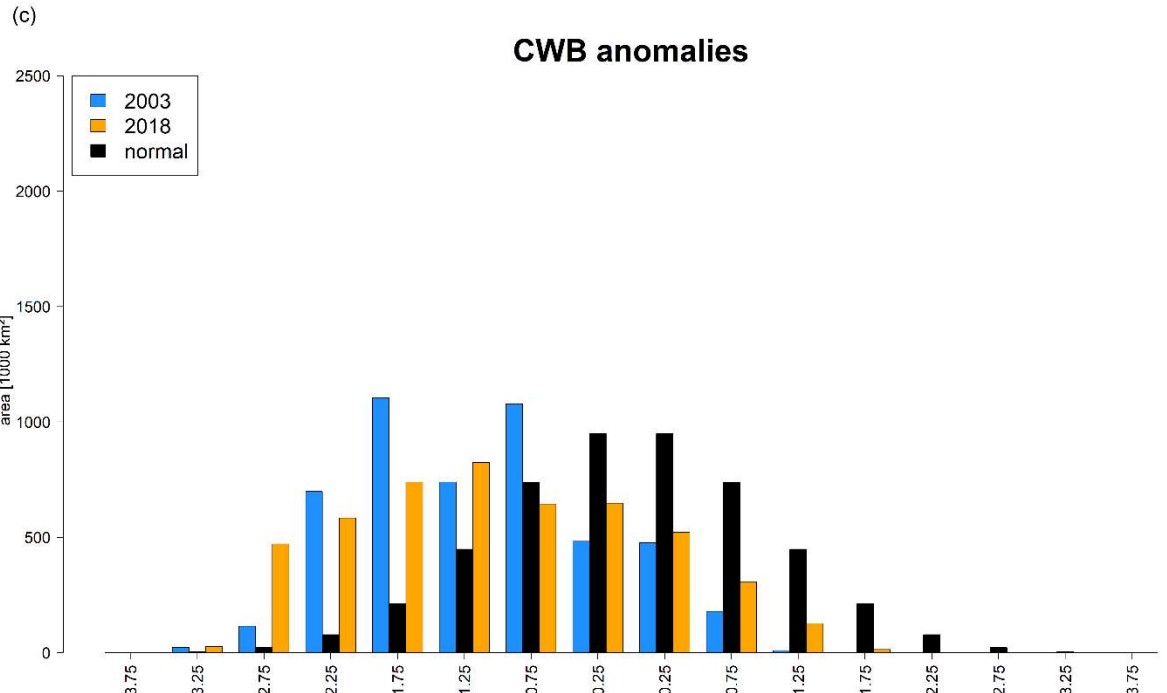

Figure 3: Maps depicting standardized anomalies of climatic water balance for August 2003 and July 2018 (a, b) as well as corresponding area histograms (in units of 1000 km²) for 2003 (blue), 2018 (orange), compared to a normal distribution (black) (c). Blue colours in (a) and (b) indicate relatively moist conditions, whereas red colours indicate dryer conditions in comparison to the 1981-2018 mean. Time-series depicting the temporal development of CWB anomalies for Northern, Central, and Southern Europe are shown in supplementary Fig. S5. A video depicting the spatio-temporal development of CWB anomalies from January through October is provided in supplementary V2 (http://doi.org/10.5446/44028).

Vegetation response – as approximated using satellite-based vegetation indices – indicated clear differences between August 2003 and July 2018. We found low NDVI quantiles in large parts of Central Europe, Southern Scandinavia, and the Baltic Sea region and high quantiles in the Mediterranean in July 2018 (Fig. 4b). In contrast, 2003 featured low NDVI quantiles in Western, Central, and Southeast Europe and high quantiles in Northern Europe (Fig. 4a). The most prominent difference between July 2018 and August 2003 was the 1.5 times larger area featuring the lowest quantile, i.e. 409,000 km² in July 2018 vs. 272,000 km² in August 2003 (Fig. 4c). At the same time, a 1.7 times larger area featured the highest quantile in July 2018, i.e. 117,000 km² vs. 68,000 km² August in 2003 (Fig. 4c). According to NDVI quantiles, hotspots of drought-response in July 2018 were located in Ireland, United Kingdom, France, Belgium, Luxemburg, the Netherlands, Northern Switzerland, Germany, Denmark, Sweden, Southern Norway, Czech Republic, Poland, Lithuania, Latvia, Estonia, and Finland.

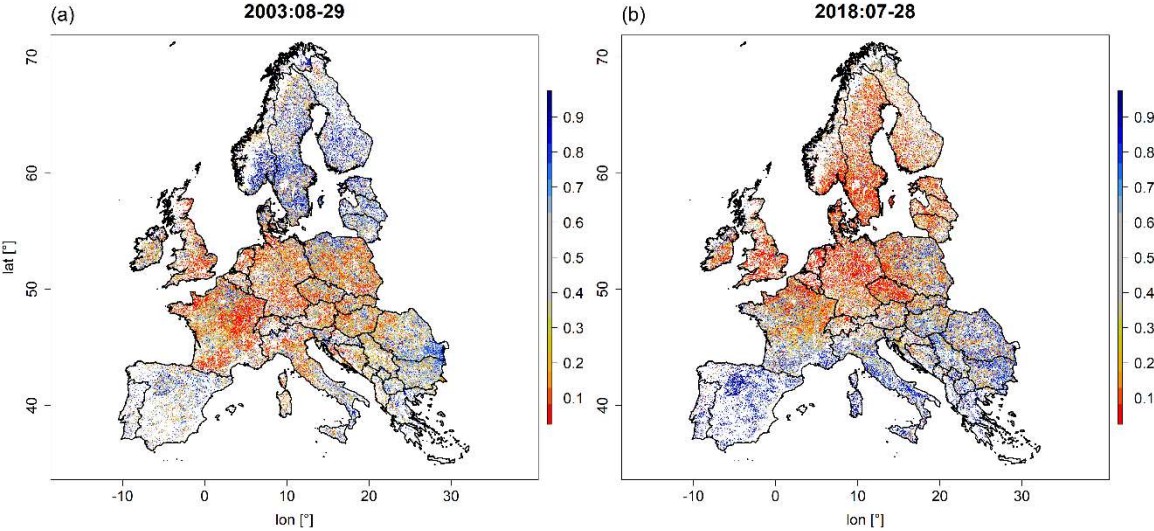

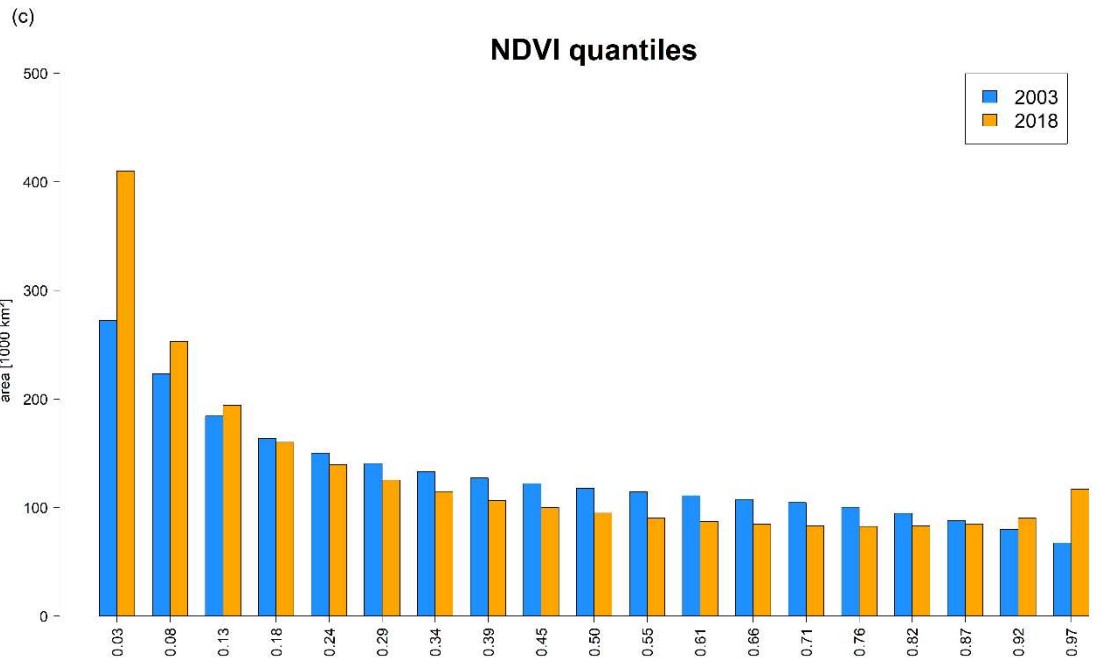

**Figure 4: MODIS NDVI quantiles representing peak of drought conditions at the end of August (DOY 241) in 2003 (a) and end of July (DOY 209) in 2018 (b) for the five selected land-cover types as well as the corresponding area histograms (in units of 1000 km²) representing the nineteen NDVI quantiles (c). Blue colours in (a) and (b) indicate upper quantiles (thus a higher than average vegetation greenness), while orange to red colours indicate lower anomalies (i.e. lower than average vegetation greenness). Blue bars in (c) refer to 2003 and orange bars to 2018. Complementary results for MODIS EVI are shown in supplementary Fig. S7. Videos depicting the spatio-temporal development of NDVI and EVI are shown in supplementary V3 and V4 (http://doi.org/10.5446/44029 and http://doi.org/10.5446/44030).**

In regions with water deficit (CWB anomalies below - 2 SD and below 0 SD; Figs. 5a, 5b, respectively) we found a higher frequency of low NDVI quantiles compared to upper quantiles. In detail, regions with extreme water deficit (5a), featured higher absolute areas with low NDVI quantiles for all land cover classes but broadleaved forest in 2018 compared to 2003. In regions with weak water deficit (5b), absolute areas were more similar between 2003 and 2018 but coniferous and mixed forests featured larger areas with low quantiles in 2018. Similarly, regions with no water deficit (CWB anomalies above 0; Fig. 5c) featured higher frequencies of upper quantiles compared to lower quantiles which however was more pronounced in 2018 for pastures, arable land, and broadleaved forests and more pronounced in 2003 for coniferous and mixed forests. The most prominent difference was related to absolutely larger areas being affected by water deficit in 2018 compared to 2003 (see also Fig. S6), particularly considering lowest VI quantiles for coniferous forests with a more than 8 times higher area in 2018 (Fig. 5a). If considering relative frequencies, histograms of 2018 and 2003 became more similar but yet revealed a higher proportion of lowest quantiles in 2018 (Fig. S8, for results based on EVI see Figs. S9-S10). This observation was confirmed when considering only pixels with extreme (CWB anomaly < -2) or moderate (-2 < CWB < 0) water deficit (Figs. S11-S13) in both 2003 and 2018. Although the differences of absolute areas decreased in this comparison, 2018 generally displayed larger areas with lowest quantiles compared to 2003. However, for EVI, subtle differences of opposite sign were observed for regions featuring extreme water deficit, while regions with moderate water deficit expressed similar patterns as for NDVI (Fig. S13 compared to Fig. S12).

The temporal development of NDVI quantiles from the corresponding drought regions (CWB anomaly < -2 in August 2003 vs. July 2018) differed between the two events (Fig. 6). While pastures featured rather similar temporal patterns of NDVI quantiles in the two years, arable land as well as the three considered forest types featured lower mean quantiles over most of the growing season and particularly in early summer. In combination, the 2018 time series featured lowest mean quantiles at the end of July, while the 2003 lowest mean quantiles occurred during August.

The impression of CWB affecting NDVI quantiles was underpinned by the linear regressions between the logit-transformed NDVI quantiles and the CWB-anomaly in 2003 and 2018, respectively (Fig. 7a-f). For all land-cover classes a significant and positive effect of climatic water balance on NDVI quantiles was observed, particularly in 2018. Explained variance ($r^2$) and bootstrapped model slopes were consistently higher in 2018 compared to 2003 (Fig. 7g). In addition, $r^2$ and model slopes were highest for pastures in both years, followed by arable land, and the three forest types which did not differ among each other in 2018 while broadleaved and mixed forests featured smaller slopes compared to coniferous forests in 2003 (lower case letters in Fig. 7g). In comparison, the differences between 2003 and 2018 were highest for pastures, followed by arable land, broadleaved forests, mixed forests, and coniferous forests. The linear mixed effects model over all land-cover classes and the two years confirmed the significant fixed effect of climatic water balance on logit-transformed NDVI quantiles (marginal $r^2$ = 0.37). Incorporation of random slopes related to land cover and year increased explained variance by 33 percent (conditional $r^2$ = 0.48), confirming the varying effect of the two drought events as well as the differing impact on different land-cover

classes. All presented results based on NDVI are generally confirmed by complementary analyses using EVI (supplementary Figs. S7 and S9-S10 and S13-S15).

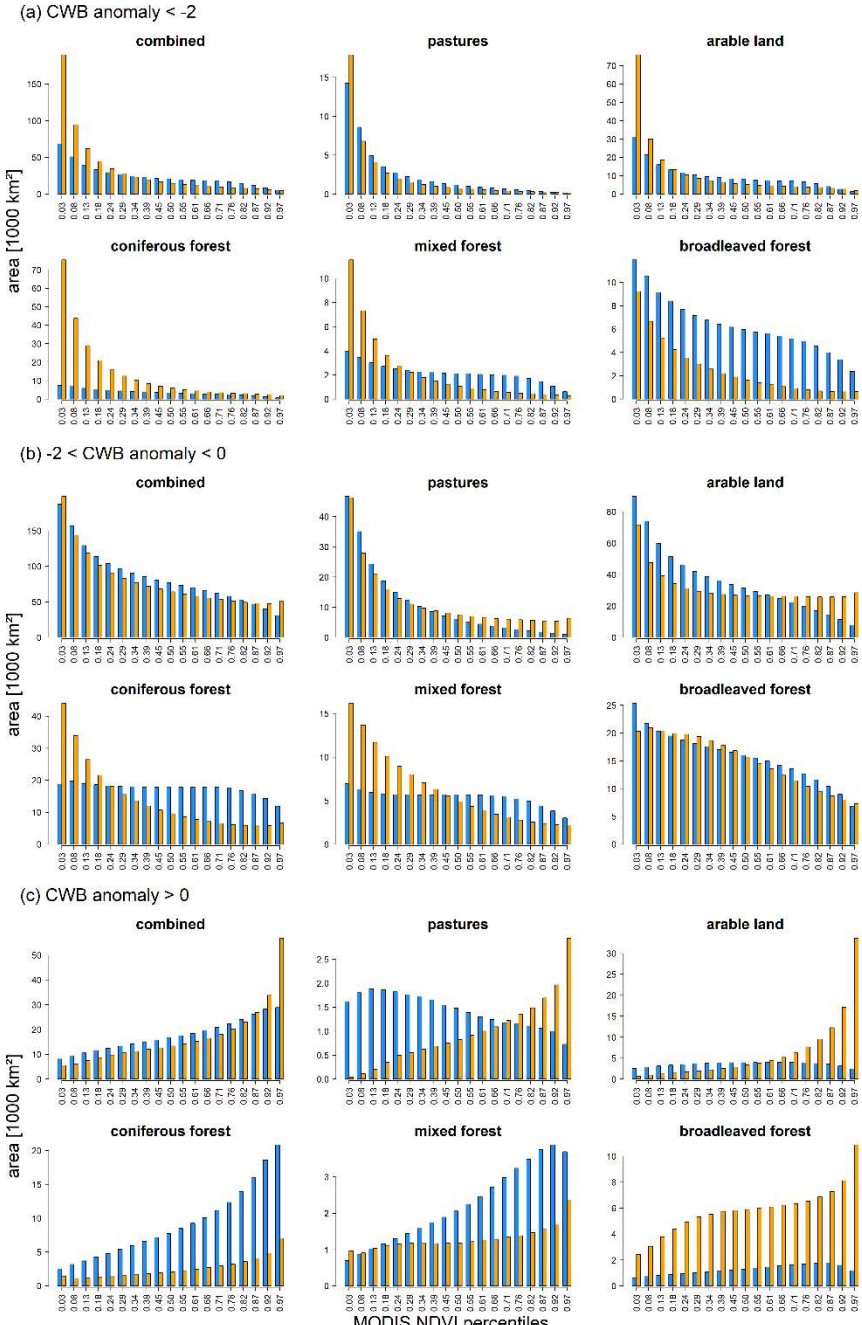


**Figure 5: Histograms depicting the absolute areas (in units of 1000 km²) representing the nineteen NDVI quantiles pooled according to CORINE land-cover classes for regions with (a) extreme water deficit (CWB-anomaly < -2) vs. (b) weak water deficit (- 2 < CWB-anomaly < 0) and (c) no water deficit (CWB-anomaly > 0). Blue bars refer to 2003, orange bars to 2018. Histograms depicting the proportional areas are shown in supplementary Fig. S8, results for MODIS EVI in supplementary Figs. S9 and S10.**

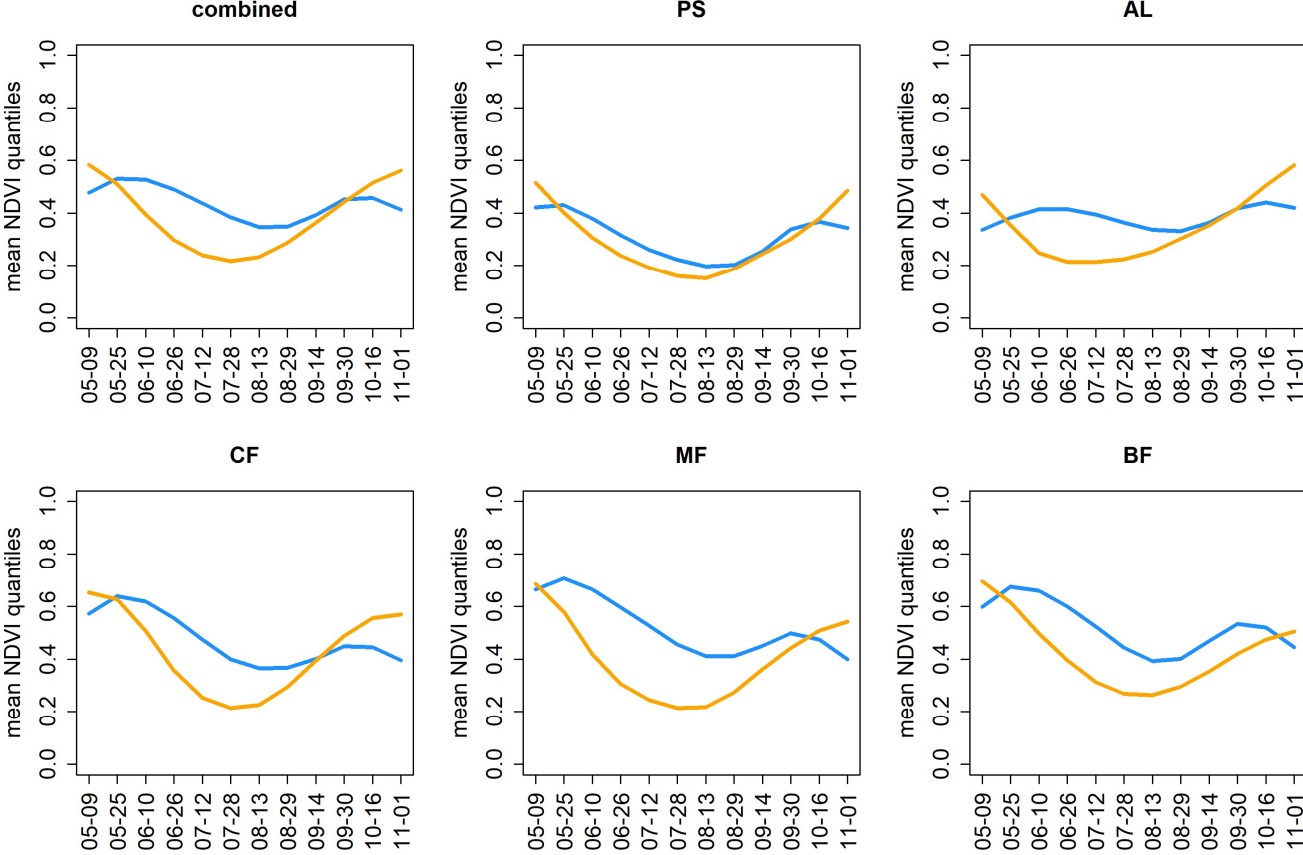


**Figure 6: Time-series of NDVI-quantiles averaged over the regions featuring CWB < -2 in August 2003 (blue) and July 2018 (orange) for the five different land-cover classes (b-f) and a combination of those (a). PS = pastures, AL = arable land, CF = coniferous forest, MF = mixed forest, BF = broadleaved forest. Complementary results for MODIS EVI are shown in supplementary Fig. S14.**

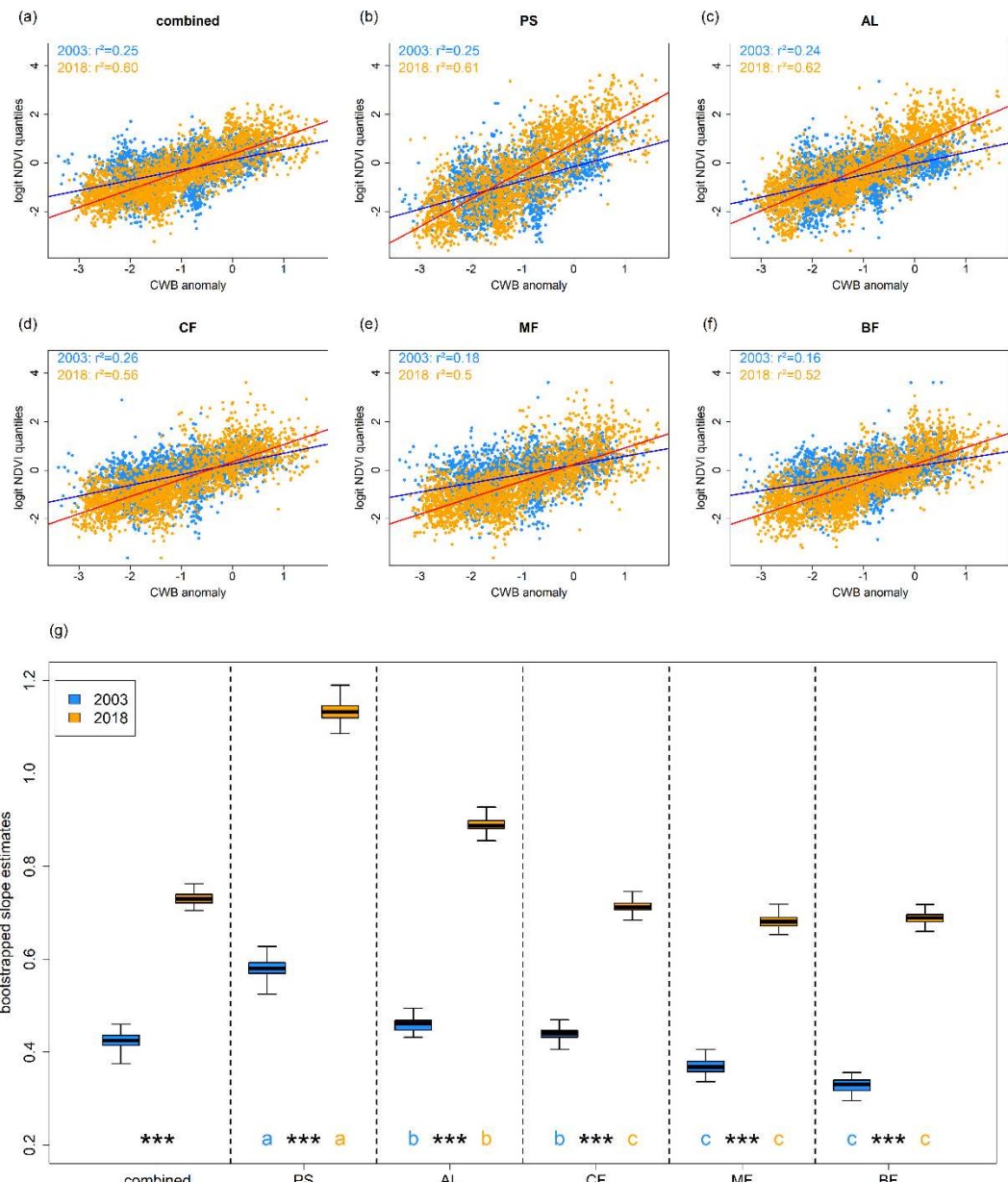

Figure 7: (a-f) Scatterplots depicting the relationship between average logit-transformed NDVI-quantiles and mean CWB in 2003 (blue) and 2018 (orange) for pastures (b), arable land (c), coniferous forests (d), mixed forest (e), broadleaved forest (f) and a combination of those (a). Blue lines depict the regression line for 2003, red lines for 2018. (g) Bootstrapped regression slope estimates for the five different land-cover classes as well as their combination. Minor case letters refer to group assignment of land-cover classes according to the overlap of 99.9 % confidence intervals of bootstrapped slopes in 2003 (blue) and 2018 (orange). Significance stars (***) indicate no overlap between 99.9 % confidence intervals of 2003 and 2018 for the respective land-cover class. PS = pastures, AL = arable land, CF = coniferous forest, MF = mixed forest, BF = broadleaved forest. Complementary results for MODIS EVI are shown in supplementary Fig. S15.

**4 Discussion**

**4.1 Climatic framework**


Based on the parameters considered for the quantification of summer conditions, 2018 clearly supersedes the record drought of 2003 (Figs. 1-3). The more extreme anticyclonic circulation patterns for 2018 across Northern Europe as indicated by 500 hPa geopotential height likely triggered stronger positive Tmax anomalies and more negative climatic water balance anomalies across Central and Northern Europe in comparison to 2003. Moreover, the two drought events differed regarding their timing


and location, with 2018 featuring an earlier peak of drought (July vs. August in 2003) and a more northward centre around the Baltic Sea vs. the Mediterranean in 2003.

Anticyclonic blocking situations – as indicated by strong positive 500 hPa geopotential height anomalies – have been reported an increasing frequency in course of the satellite era (Horton et al., 2015) which likely relates to the increasing persistence of heatwaves observed over the past 60 years (Pfleiderer and Coumou, 2018) and the increasing frequency of a hemisphere-wide


wavenumber 7 circulation pattern (Kornhuber et al., 2019). The resulting heatwaves are additionally enhanced by global warming and positive land surface – atmosphere feedback loops via soil moisture depletion and subsequent lack of latent cooling (Fischer et al., 2007). Moreover, summer temperatures and precipitation were reported to correlate negatively at mid latitudes which may amplify further according to CMIP5 climate projections (Zscheischler and Seneviratne, 2017). This renders the evaluation of ecosystem responses to compound events a key topic for climate change research (Zscheischler et


al., 2018). Concluding, the prevailing extreme climatic conditions qualify 2018 as a key event for studying ecosystem-responses to hotter droughts in Europe.

**4.2 Ecosystem impact**

**4.2.1 Stronger ecosystem impact in 2018**


The comparison of MODIS NDVI anomalies generally supported the impression of a more extreme drought in July 2018 compared to August 2003. That is, the area featuring the lowest quantile in July 2018 was 1.5 times as high compared to August 2003 (Fig. 4c). In accordance with the more northward location of the anticyclone, hotspots of negative ecosystem impact were concentrated in Central and Northern Europe in 2018. Our analyses based on EVI generally confirmed these observations (Figs. S7, S9-S10 and S13-S15).


In general, the spatial distribution of NDVI quantiles matched the observed climatic dipole of 2018 very well. Moreover, the bimodal response of European ecosystems is well in line with a report on European maize yield in 2018, with an observed strong increase (10% more than average) for e.g. Romania and Hungary, a strong decrease (10% less than average) for e.g., Germany and Belgium, and the European level net effect estimated at a decrease in maize yield of around 6% (see public references: European Commission).


The bimodal behaviour and larger extent of ecosystem impact in 2018 was also reflected in land-cover specific area distributions of NDVI quantiles (Figs. 5a and S6). That is, the area featuring lowest quantiles in regions with extreme water

deficit was much larger in July 2018 compared to August 2003. Given the skewed distribution of VI quantiles in those regions, i.e. more lower quantiles in regions with extreme negative CWB anomalies and more upper VI quantiles in regions with positive CWB anomalies, the response of these ecosystems appears to be governed by prevailing climate conditions, in particular climatic water balance.

This impression was underlined by linear regressions, which revealed a strong and significant positive impact of CWB on NDVI quantiles in both years and for all land-cover types (Figs. 7a-f). It is noteworthy, that 2018 was characterized by a higher explained variance and significantly more positive model slopes in comparison to 2003 (Fig. 7g). These impressions from single linear models were generally supported by the linear mixed effects model, which showed an increase of $r^2$ at about 33 percent when incorporating land-cover and year as random effects.

We want to note that we also observed small areas (ca. 1000 km² for each of the considered land cover classes) of high quantiles in regions featuring extreme drought. This observation is likely related to the different resolution of climate data (0.5°, thus ca. 50 km x 50 km) vs. MODIS (231 m x 231 m). Using such relatively coarse climate data for quantifying drought neglects elevation-driven climatic variations within grid cells (Zang et al., 2019) in contrast to the relatively higher resolution of MODIS which likely captures such differences. For instance, mountain ranges – which on average feature cooler temperatures (thus less evapotranspiration) and higher precipitation – likely feature less extreme water deficit compared to surrounding lowlands which may cause higher VI values than one would expect if only considering the drought classification of the corresponding coarsely resolved climate grid cell. In addition, groundwater availability – which may vary within climate grid cells, particularly in proximity to rivers and lakes – may locally modulate plant water availability, possibly explaining some of the observations of high VI quantiles under extreme drought.

Furthermore, quantifying drought on anomalies alone bears the risk to erroneously classify regions that actually feature water surplus (CWB > 0) as being drought-affected (Zang et al., 2019). This is likely to happen in regions which usually have high water surplus (e.g. the Norwegian west-coast) and thus may feature more than two negative CWB standard deviations even though raw CWB is positive (Zang et al., 2019). For both 2003 and 2018, 5 (1) percent of climate grid cells featured positive CWB even though CWB anomalies indicated (extreme) water deficit (CWB < 0 and CWB < -2 SD, for details see Fig. S16). This false classification may explain some of the observed highest quantiles in regions that were classified as featuring extreme water deficit. Lacking a more advanced drought-metric that compensates for such effects, we followed the widely accepted approach to use standardized CWB for our analyses. To show the full picture, we point out potential biases caused by this approach (Fig. S16) as proposed by Zang et al. (2019). Given the relatively low number of potentially misleading extreme drought classifications (1 percent for each year) we estimate the impact on our analyses to be generally low.

Finally, we want to stress that the choice of VI used for quantification of the ecosystem response to drought matters. While Li et al. (2010) found stronger relationships and lower errors between NDVI and ground observations made in grasslands, shrublands and forest in comparison to EVI, Vicca et al. (2016) reported EVI to be more sensitive to reductions in GPP that were not reflected by leaf coloration or early leaf senescence. Consequently, the choice of VI may also affect the differences in the responses of different vegetation types. Therefore, we present results from both VIs which generally support each other

(e.g. comparison between Figs. 4 and S7, Figs. 7 and S15) but focused on NDVI to make our contribution directly comparable to previous studies dealing with drought impacts on ecosystems (Anyamba and Tucker, 2012; Orth et al., 2016; Xu et al., 2011). To complement our VI-based assessment, future studies may consider the use of solar-induced fluorescence which is known to be more sensitive to terrestrial photosynthesis (Li et al., 2018).


### 4.2.2 Location and timing of drought possibly drives stronger ecosystem response in 2018

The stronger coupling (higher r²) between CWB and VI quantiles in July 2018 may be related to the period used to integrate CWB (4 months), namely if the VI response of August 2003 were triggered by climatic anomalies over longer or shorter periods. To test this, we assessed the relationship of VI quantiles with CWB integrated over various periods (e.g. including

previous winter in CWB or shortening the period) which all revealed similar patterns, i.e. a stronger response to CWB in 2018 (not shown).

However, it seems possible that the stronger coupling of VI quantiles to CWB in 2018 is related to the earlier timing and more northward location of drought and thus different ecosystems being affected at an earlier stage of their seasonal cycle. In 2003, the centre of drought was in Central and Southern Europe and peaked in August, i.e. it affected regions which host ecosystems

that are frequently experiencing summer drought at that time of the year and thus are likely better adapted to cope with dry conditions. In contrast, the circulation patterns of 2018 resulted in a drought-centre in Central and Northern Europe earlier in summer, i.e. in regions with ecosystems that are less adapted to extremely dry climatic conditions and therefore likely react strongly. The earlier timing possibly also triggered a stronger response to the drought in 2018, particularly in Northern Europe where it began as early as May, i.e. at the beginning of the growing season (Fig. S5 and supplementary video V2). Northern

European forests are dominated by coniferous forests that to a large degree consist of Norway spruce and Scots pine, i.e. two tree species that have been frequently reported to suffer under drought (e.g. Buras et al., 2018; Kohler et al., 2010; Rehschuh et al., 2017; Rigling et al., 2013). Moreover, coniferous forests provided a high share of the drought affected ecosystems in 2018 (Fig. S6). In combination, the potentially stronger reaction of high latitude coniferous forests may partly explain the observed stronger coupling between CWB anomalies and VI quantiles in 2018 compared to 2003. In addition, various reports

from dried-out pastures and cornfields as well as deciduous trees shedding their leaves in July and August in 2018 likely explain the record-low VI values for corresponding Central European ecosystems (see public news references and Fig. S17 for an example of early leaf senescence of European beech in 2018). At the same time, the usually summer-dry Mediterranean experienced water surplus, relaxing the general limitation of the associated ecosystems by plant water availability and leading to a generally higher greenness of the vegetation. Considering forest ecosystems, this interpretation is in line with Klein (2014),

who reported higher leaf gas exchange and thus photosynthetic capacity under dry conditions in Mediterranean forests compared to temperate forests.

Nevertheless, our hypotheses explaining the stronger coupling of VI quantiles with CWB in 2018 need further support, e.g. by studying the sensitivity (model slope) and coupling (r²) of plant productivity with climatic properties for the considered land-cover classes as for instance done by (Anderegg et al., 2018). A sub-classification of land-cover classes seems to be reasonable

for such an analysis, in order to account for possibly differing drought-sensitivities of ecosystems represented by one specific land-cover class. For instance, Mediterranean coniferous forests comprise different species and are thus likely better adapted to drought than boreal coniferous forests in Scandinavia.

As a first step into this direction, we only compared VI-quantiles of regions that featured extreme or moderate water deficit in both years, thus only considering regions representative of the exactly same ecosystems. On average, we again observed a higher share of low quantiles in 2018 compared to 2003 (Figs. S12 and S13). Only for the EVI in regions featuring extreme drought, 2003 featured a slightly higher share of lower quantiles compared to 2018. Taken together, it nevertheless seems that even when only considering the same regions for the comparison between both years, the impact of the 2018-drought supersedes the one of 2003. Interestingly, in this comparison clear differences were only observed for forest ecosystems. This might indicate so-called drought legacy effects (see also section 4.2.4) as a consequence of preceding extreme droughts, such as the one of 2015 after which an increased forest mortality and growth decline was observed in southern Germany and other parts of Central Europe (Buras et al., 2018).

### 4.2.3 Forests featured a weaker immediate response to drought

In addition to the differences between the two drought events, we also observed differing sensitivities (model slopes) of VI quantiles to CWB among ecosystems, which were consistent over the two drought events. We found the highest regression slope estimates for pastures followed by arable land and forests (Fig. 7g). This likely reflects the higher climatic buffering function of forests in comparison to arable land and pastures. In forests, the micro-climate is generally less extreme, leading to lower ambient air temperatures and consequently a lower evapotranspiration in comparison to open fields (Chen et al., 1993, 1999; Young and Mitchell, 1994). Consequently, water resources are consumed more sustainably by forests. Moreover, if not growing on water-logged soils trees typically feature higher rooting depths compared to grasses and crops and therefore have access to deeper soil water reservoirs. Regarding the European drought of 2003, an accelerated soil moisture depletion of grasslands in comparison to forests has been reported earlier (Teuling et al., 2010). Also Wolf et al. (2013) found contrasting responses of grasslands vs. forests regarding water and carbon fluxes during a drought event in 2011. They observed an immediate negative drought-impact on the productivity of managed grasslands while mixed and coniferous forests simply reduced transpiration and maintained GPP, thereby increasing their water-use efficiency and decreasing soil-water consumption (Wolf et al., 2013). As mentioned in section 4.2.1 the choice of VI may alter the differences in the responses of different vegetation types to drought. However, the fact that NDVI and EVI indicated differences of similar sign and magnitude among the considered ecosystems, supports our interpretation. Nevertheless, further remote sensing products such as the solar-induced fluorescence may provide additional information on ecosystem-specific drought responses.

### 4.2.4 Forest legacy effects

Despite an observed lower sensitivity, European forests were in parts heavily affected by the drought 2018, as indicated by the distribution of VI quantiles (Fig. 5a). For instance, 180,000 km² of drought affected (CWB < 0) coniferous forests featured the

lowest quantile in 2018. Moreover, 45,000 km² of deciduous trees featured the lowest VI quantile in 2018 which likely relates

to the observed early leaf-shedding of deciduous trees in Central Europe (see Fig. S17). First estimates by German forestry about the total loss of wood volume as a consequence of the drought 2018 and the ongoing drought in 2019 were in the order of 105 million solid cubic meters (see public references, BMEL). Besides these direct impacts, carry-over effects are likely to be experienced in the following years: Evidence for delayed responses comes from remotely sensed vegetation activity in the aftermath of the 2003 event (Reichstein et al., 2007) and reports on observed dieback of Norway spruce, Scots pine, and

European beech in spring and summer 2019 (GfÖ-workshop on drought 2018 held on June 4th, 2019 in Basel, see also public news references). These effects could partly be due to legacy effects in tree response to drought (Anderegg et al., 2015; Kannenberg et al., 2018) as well as tree mortality often occurring years after the event (Bigler et al., 2006; Cailleret et al., 2017). Support for an expected delayed response of forests also comes from a severe drought in Franconia, Germany, in 2015, which resulted in increased Scots pine mortality, that was not recognized earlier than in the subsequent winter 2015/2016 and

became even more pronounced in spring 2016 (Buras et al., 2018). From this event, we also learned that particularly forest edges – which feature an intermediate micro-climate between the forest interior and open fields – are more susceptible to drought-induced mortality (Buras et al., 2018). However, given the spatial resolution of the applied remote sensing products (231 m x 231 m), patches with tree dieback as well as forest edges could not be resolved, which should therefore be given attention in future studies.


### 4.3 Call for a European forest monitoring

Given likely legacy effects, studying the development of Central and Northern European forest ecosystems over the next years is particularly interesting and may reveal negative mid- to long-term responses as well as an increased tree mortality. In this context, we propose an observation of forests within the outlined hotspots by combining satellite-based and close-range remote

sensing techniques with dendroecological investigations and an eco-physiological monitoring (Buras et al., 2018; Ježík et al., 2015). The initiation of such monitoring campaigns would provide the unique opportunity to study natural tree die-back in real-time, thereby increasing our knowledge about drought-induced tree mortality (Cailleret et al., 2017). The recently released European forest condition monitor (www.waldzustandsmonitor.de) may be considered a first step towards a satellite-based, near real-time monitoring of European forests.


### 5. Conclusions

Based on climatic evidence, 2018 may be considered the new reference year for hotter droughts in Europe. However, the observed contrasting spatiotemporal patterns of the drought events in 2003 and 2018 highlight the complexity of ecosystem responses to severe droughts. More specifically, we observed a different sensitivity of ecosystems to CWB between the two

events and a differing sensitivity of land cover classes to drought, with pastures and agricultural fields expressing a higher sensitivity in comparison to forests. The differing sensitivity was probably caused by the differing spatial extent of the two events thereby affecting presumably less drought-resistant ecosystems in 2018. The observed climatic heterogeneity and

resulting ecosystem response poses certain challenges for estimating the effects on the carbon cycle in European ecosystems in 2018. That is, the observed dipole in 2018 makes it difficult to directly compare the European carbon budget of 2018 with 2003. Finally, in addition to quantifying impacts of the drought 2018 on European ecosystems our results possibly mirror forest drought legacies from preceding drought events. Moreover, additional legacy effects of forest ecosystems are likely to occur in course of the next years. Consequently, to obtain a more complete picture about the impact of the drought 2018 we recommend continued satellite-based remote sensing surveys to be accompanied by immediate in-situ monitoring campaigns. In this context, particular attention should be given to the outlined hotspots of the drought 2018, i.e. Ireland, United Kingdom, France, Belgium, Luxemburg, the Netherlands, Northern Switzerland, Germany, Denmark, Sweden, Southern Norway, Czech Republic, Poland, Lithuania, Latvia, Estonia, and Finnland.

**Data Availability**

The datasets analysed for this study are publicly available. Public web-links as well as the post-processing of data is described in the material and methods section of this article, wherefore all results are reproducible.

**Author Contributions**

AB and CSZ developed the study design. AB conducted all data processing and statistical analyses. Interpretation and refinement of statistical results was discussed among AB, AR, and CSZ. AB drafted the first version of the article which was further refined by AR and CSZ.

**Competing Interests**

The authors declare that the research was conducted in the absence of any commercial or financial relationships that could be construed as a potential conflict of interest.

**Acknowledgements**

This project is funded by the Bavarian Ministry of Science and the Arts in the context of the Bavarian Climate Research Network (BayKliF). We acknowledge the valuable suggestions made by two anonymous reviewers which made us improve our study.

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
