# Peer review of "Quantifying impacts of the drought 2018 on European ecosystems in comparison to 2003"

_Biogeosciences, 2019_

## Referee Comment (RC1) · Anonymous Referee #1 · 19 Aug 2019

In this manuscript, the authors compare the climatological features of two intense drought events over Europe, the 2003 and 2018 heatwaves. From a climatological analysis, they carry on to analyze the effect of both events on European vegetation. The authors' results are based on a suite of statistical analyses of MODIS-based vegetation indices combined with a widely used land cover map. Their main conclusion is that the 2018 heatwave had a stronger effect on European vegetation than the 2003 one.

General comments Overall the manuscript is well written and follows a logic questioning line going from the climatology of the heatwave to the effect on vegetation. However, I am not convinced by the relevancy of the way the main question is addressed. From the first climatological data shown by the authors it is obvious that the 2003 and 2018

events are very different in terms of location (baltic countries in 2018 versus central Europe in 2003) and timing (july was the beginning of the 2003 heatwave whereas it was the end of the 2018 one) and, even though it is not shown, in initial conditions. These crucial differences are however mostly ignored in the way the analyses are designed, potentially pointing to severe flaws in the results, that I detail below. This observation leads me to suggest more detailed analyses be carried out before publication.

Detailed comments Effect of heatwave timing, duration and legacy The analyses carried out, only encompassing the greenness index in july of both years, seem too superficial to draw the far-reaching conclusions the authors make. First on heatwave timing, the stronger NDVI signal in july 2018 compared to july 2003 could be the result of the difference in timing. This point is ignored when concluding that the 2018 heatwave had a stronger effect on European ecosystems than the 2003 one. Second, as mentioned as a discussion item, the legacy of water balance can be very important for heatwave effects on forests especially. Even though this is be a major point underlying the relevancy of the question asked, no data or analysis shown tries to compare the water conditions prior to both heatwaves. An analysis of both heatwaves time evolution and the comparison of each heatwave's end month, that could then be different from one to the other could be an option.

Heatwave location defining the ecosystems affected Another point that undermines the results presented is the comparison of vegetation types in absolute terms without a prior description of vegetation types affected in each case that might be very different. Even though this information is essential to make sense of a comparison of the effects on vegetation of two climatic events, it is hardly discussed and made very hard to see in the way the data are processed. For example the varying y-axis ranges in figure 5 hide the relative weights of each vegetation type with water deficit or water surplus. Figure S3 might be more explicit to this regard by considering relative vegetation cover.

Statistical indices Finally, another aspect of the manuscript that makes it less convincing is the choice of the figures and complex statistical indices derived. For example

figure 5 is hard to interprete. What is the implication of high NDVIs combined with low CWB? In general, the methods section is very concise, making it easy to read but also lacking some key points to help the reader understand the many indices used. For example even though they are widely used NDVI and EVI should be defined. Also, heat load variable is not defined and is sometimes written heat load and sometimes heat-load. If it is simply Tmax call it this.

―――――――――――――――――

---

## Referee Comment (RC2) · Anonymous Referee #2 · 9 Sep 2019

This is a well-written study comparing the European heatwaves of 2003 and 2018. Comparison of climatological data and vegetation indices lead to the conclusion that the 2018 heatwave was more severe than the 2003 heatwave. However, substantial regional differences occur. The idea behind this study is interesting, but the study misses out on several aspects needed to support the conclusions. Especially the lack of the temporal patterns in weather data makes it hard to evaluate the results. No time series for temperature, precipitation or drought indices are shown to illustrate the heatwave patterns of both years. Moreover, end of July was chosen as the study period for the impact on vegetation, hence ignoring any change that took place in August (e.g. the massive forest fires in Portugal early August 2018).

Specific comments: l.96ff: I suggest to simply argue that you focused on March-Oct

because that is the period of interest for vegetation dynamics and leaving out winter helps avoiding artefacts (e.g. snow cover, but also defoliation in deciduous systems). l104-105: interpolation may create artefacts when searching for anomalies – especially when there are gaps during the drought episode under study. I suppose this is of minor importance for this study, because gaps are less likely during periods of drought (i.e., no clouds), but I wonder if the interpolation can be avoided. If not, the possibility for such artefacts should at least be discussed. l.119ff: NDVI and EVI are mainly greenness indicators. They may reflect photosynthesis, but not if photosynthesis changes without changing greenness. This is particularly relevant for drought. In this sense, EVI is better than NDVI (see Vicca et al 2016, Scientific Reports). I therefore advise to use the EVI results rather than the NDVI in this manuscript. It should also be clearly indicated what these VIs can reflect (and what not!). This is completely missing from the discussion of the current manuscript, but needs to be discussed (i.e., are we looking at green biomass/browning/defoliation. . . and what are the implications for e.g. legacy effects). l.141-142: awkward phrasing: Subsequently, we for 2003 and 2018 determined . . . should be: Subsequently, we determined the difference between 2003 and 2018 for the respective metric. . . l.149ff: the timing of the heatwaves should be demonstrated with data to justify the choice to focus on end of July. Time series of temperature and CWB for e.g. France, Germany (which suffered from the heat in both 2003 and 2018), or even for the different regions (N, W, S, Central Europe). How sensitive is your analysis to the time choice? Are results similar if the analyses were repeated for end of August for example? l.150-151: VIs cannot be lower than 0. (anomalies can) l.159: What was done with pixels where land cover changed between 2003 and 2018? Was that even considered? (I don't think it will have a big impact on the analyses, but it's worth a mention). l.191: 0.55 should be 55% I suppose. Fig.1: why was the timing April-July chosen for these figure? This is not motivated in the text I think. A time series with weather data would be very helpful to evaluate this choice (see earlier comment). I noticed that this is briefly mentioned in the discussion (l. 320), but data are not shown. Please do show these data. Fig.4: It is unclear where VI-deviations

from the mean were significant. Please clarify, also in the text. l.240ff: A map with vegetation types is missing to illustrate where the different vegetation types occur and how the differences in impact for the different vegetation types correspond with the regional differences (e.g. Scandinavia being dominated by conifer forests). Fig.6: consider moving to appendix, adding instead a figure with time series for weather data. l.251: Fig. 7 shows EVI, not NDVI. The text is about NDVI. (I suggest to focus on EVI for in the main document and move NDVI to appendix – see earlier comment). l.328ff: Portugal suffered from severe wildfires in 2018. This is not included in the analyses because the fires occurred mostly in August and the analyses are only for April-July. Other important events may be missed out because of the choice for April-July. l.340ff: I suggest to include in this part of the discussion some text on the relationship between ecosystem types and climatic regions and how this may/may not influence the interpretation (see also earlier suggestion for figure addition). l.358: public news references are not appropriate for this statement.

———————————————————

---

## Author Comment (AC1) · 30 Sep 2019

Thank you very much for your constructive review, which highlights important aspects and potential ambiguities and will help us to considerably improve our manuscript.

While we agree with most of your critiques, we would like to emphasize that we do not ignore the differences of the two drought events: Fig. 6 was particularly designed to highlight that the peak of the drought-response in different land-cover types was delayed in 2003. As shown in Fig. 6 the lowest value of mean NDVI-quantiles of coniferous and mixed forests was lower in 2018 compared to 2003, while broadleaved forests displayed more similar NDVI-quantiles. Regarding arable land and pastures the lowest values occurred later in time and were lower in 2003. We refer to these results in

lines 247-250 in the initial submission. Moreover, section 4.2.2 of the initial submission discusses possible effects related to the differing location, i.e. the drought 2018 hit potentially less adapted ecosystems at higher latitudes which may have caused the observed stronger ecosystem response in 2018.

We agree, that these points need more emphasis and we will do so in the revision of our manuscript by:

1) Visualizing the temporal development of climatic conditions in a similar manner as for NDVI in Fig 6 in the main text and adding an animated gif to the supplementary which depicts the spatiotemporal development of climatic water balance anomalies (i.e. an animated version of Fig. 3 from January to October) to quantify the preceding initial conditions of the two events (moist winter and early spring in 2018 thus beneficial conditions at the onset of 2018 vs. dry late winter in 2003 thus less beneficial initial conditions in 2003) and the development in course of the drought events,

2) Providing yet another supplementary gif which will visualize the temporal development of NDVI quantiles using maps and histograms (animated version of Fig. 4 throughout the growing season) from beginning of May until end of October to allow for assessing the temporal development of the two drought events,

3) Comparing and modelling the climatic features and ecosystem response representative of the peak of drought in each year (i.e. July/DOY 209 in 2018 vs. Aug/DOY 241 in 2003) instead of for July/DOY 209 only and revise the underlying analyses of Figs. 4, 5, and 7 accordingly,

4) Emphasizing the spatiotemporal effects in the discussion, in section 4.2.2.

As can be seen in the figures attached to this reply, these additional analyses confirm our initial conclusion since:

I) The area featuring extreme drought (lower than 2 negative standard deviations) was larger in 2018 compared to 2003 (1.4 times larger, i.e. 1.35 million km$^2$ in 2018 vs.

950,000 km$^2$ in 2003).

II) The drought response of the considered ecosystems affected a larger area in 2018 compared to 2003 (again 1.4 times larger, i.e. 820,000 km$^2$ featuring the lowest quantile in 2018 and 570,000 km$^2$ in 2003).

III) The drought response was stronger in 2018 compared to 2003 as expressed by significantly steeper regression slopes, the differences between 2003 and 2018 now even became stronger.

Please also find a detailed point by point reply to the comments raised by referee #1 in the supplementary pdf.

Please also note the supplement to this comment: https://www.biogeosciences-discuss.net/bg-2019-286/bg-2019-286-AC1- supplement.pdf

―――――――――――――――――――――

[Figure]

[Figure]

**Fig. 1.**

[Figure]

[Figure]

**Fig. 2.**

[Figure]

**Fig. 3.**

**Supplement:**

In this manuscript, the authors compare the climatological features of two intense
drought events over Europe, the 2003 and 2018 heatwaves. From a climatological
analysis, they carry on to analyze the effect of both events on European vegetation.
The authors' results are based on a suite of statistical analyses of MODIS-based vegetation
indices combined with a widely used land cover map. Their main conclusion is
that the 2018 heatwave had a stronger effect on European vegetation than the 2003
one.

**General comments**
Overall the manuscript is well written and follows a logic questioning
line going from the climatology of the heatwave to the effect on vegetation. However,
I am not convinced by the relevancy of the way the main question is addressed. From
the first climatological data shown by the authors it is obvious that the 2003 and 2018
events are very different in terms of location (baltic countries in 2018 versus central Europe
in 2003) and timing (july was the beginning of the 2003 heatwave whereas it was
the end of the 2018 one) and, even though it is not shown, in initial conditions. These
crucial differences are however mostly ignored in the way the analyses are designed,
potentially pointing to severe flaws in the results, that I detail below. This observation
leads me to suggest more detailed analyses be carried out before publication.

*Thank you very much for your constructive review, which highlights important aspects and
potential ambiguities and will help us to considerably improve our manuscript. While we agree
with most of your critiques, we would like to emphasize that we do not ignore the differences of
the two drought events: Fig. 6 was particularly designed to highlight that the peak of the
drought-response in different land-cover types was delayed in 2003. As shown in Fig. 6 the
lowest value of mean NDVI-quantiles of coniferous and mixed forests was lower in 2018
compared to 2003, while broadleaved forests displayed more similar NDVI-quantiles.
Regarding arable land and pastures the lowest values occurred later in time and were lower in
2003. We refer to these results in lines 247-250 in the initial submission. Moreover, section
4.2.2 of the initial submission discusses possible effects related to the differing location, i.e. the
drought 2018 hit potentially less adapted ecosystems at higher latitudes which may have caused
the observed stronger ecosystem response in 2018. We agree, that these points need more
emphasis and we will do so in the revision of our manuscript by:*

1) *Visualizing the temporal development of climatic conditions in a similar manner as for
NDVI in Fig 6 in the main text and adding an animated gif to the supplementary which
depicts the spatiotemporal development of climatic water balance anomalies (i.e. an
animated version of Fig. 3 from January to October) to quantify the preceding initial
conditions of the two events (moist winter and early spring in 2018 thus beneficial
conditions at the onset of 2018 vs. dry late winter in 2003 thus less beneficial initial
conditions in 2003) and the development in course of the drought events,*
2) *Providing yet another supplementary gif which will visualize the temporal development
of NDVI quantiles using maps and histograms (animated version of Fig. 4 throughout
the growing season) from beginning of May until end of October to allow for assessing
the temporal development of the two drought events,*
3) *Comparing and modelling the climatic features and ecosystem response representative
of the peak of drought in each year (i.e. July/DOY 209 in 2018 vs. Aug/DOY 241 in
2003) instead of for July/DOY 209 only and revise the underlying analyses of Figs. 4,
5, and 7 accordingly.*

4) *Emphasizing the spatiotemporal effects in the discussion, in section 4.2.2.*

*As can be seen in the figures attached to this reply, these additional analyses confirm our initial conclusion since:*

I) *The area featuring extreme drought (lower than 2 negative standard deviations) was larger in 2018 compared to 2003 (1.4 times larger, i.e. 1.35 million km² in 2018 vs. 950,000 km² in 2003).*

II) *The drought response of the considered ecosystems affected a larger area in 2018 compared to 2003 (again 1.4 times larger, i.e. 820,000 km² featuring the lowest quantile in 2018 and 570,000 km² in 2003).*

III) *The drought response was stronger in 2018 compared to 2003 as expressed by significantly steeper regression slopes, the differences between 2003 and 2018 now even became stronger.*

*Please also find a detailed point by point reply to the comments raised by referee #1 in the supplementary pdf.*

**Detailed comments**
**Effect of heatwave timing, duration and legacy**
The analyses carried out, only encompassing the greenness index in july of both years, seem too superficial to draw the far-reaching conclusions the authors make. First on heatwave timing, the stronger NDVI signal in july 2018 compared to july 2003 could be the result of the difference in timing. This point is ignored when concluding that the 2018 heatwave had a stronger effect on European ecosystems than the 2003 one.

*We agree that the different timing of the drought and heatwaves likely resulted in a different timing of vegetation response, as was shown already in Fig. 6 and mentioned in lines 247-250. To better visualize the observed temporal development, we will add a gif showing the temporal development of the climatic conditions, as mentioned in the reply to the general comment. We will also add supplementary gifs which depict the temporal development of NDVI from beginning of May until end of October (as can be seen here: http://www.lsai.wzw.tum.de/index.php?id=77 for 2003 and http://www.lsai.wzw.tum.de/index.php?id=75 for 2018). Moreover, we will add time-series comparable to those shown in Fig. 6 for CWB to provide a direct comparison between the climatic features of the drought affected areas in 2003 and 2018. We agree, that the temporal differences need to be pointed out more clearly, wherefore we – next to the additional analyses - will add a corresponding paragraph to the discussion where we discuss the additional analyses and the importance of drought timing.*

Second, as mentioned as a discussion item, the legacy of water balance can be very important for heatwave effects on forests especially. Even though this is be a major point underlying the relevancy of the question asked, no data or analysis shown tries to compare the water conditions prior to both heatwaves. An analysis of both heatwaves time evolution and the comparison of each heatwave's end month, that could then be different from one to the other could be an option.

*We agree that the water balance legacy effects are relevant, wherefore we will add according analyses to the manuscript. In particular, we will visualize the development of integrated CWB over time, i.e. including preceding winter conditions and extending until the end of the growing*

*season (as done for NDVI in Fig. 6). We acknowledge your suggestion to compare different months with each other. We will compare VI quantiles representing DOY 241 in 2003 and DOY 209 in 2018 in Figs. 4 and 5, and use the corresponding VI-quantiles and CWB for the modelling exercise in Fig. 7 and the mixed model. Please see the attached figures for an impression on how Figs. 4 and 7 change due to these analyses amendments.*

**Heatwave location defining the ecosystems affected**

Another point that undermines the results presented is the comparison of vegetation types in absolute terms without a prior description of vegetation types affected in each case that might be very different. Even though this information is essential to make sense of a comparison of the effects on vegetation of two climatic events, it is hardly discussed and made very hard to see in the way the data are processed. For example the varying y-axis ranges in figure 5 hide the relative weights of each vegetation type with water deficit or water surplus. Figure S3 might be more explicit to this regard by considering relative vegetation cover.

*Maybe we misunderstood your comment, but we struggle to reproduce your statement that the effects of the different locations of the drought are 'hardly discussed'. Section 4.2.2 particularly deals with the possible effect of the differing drought epicenters, where we highlight that different ecosystems' adaptation to drought likely caused the stronger ecosystem response in 2018. To account for your critique, we will add a paragraph to the corresponding methods section, highlighting the fact that the drought-affected areas differ between 2003 and 2018, which leads to varying absolute areas as well as different vegetation types that were affected. To make the picture more complete and readers aware of the uneven distribution of areas affected by the severe drought in 2003 and 2018, we will moreover add a figure to the supplementary which depicts the absolute spatial contributions of the selected land-cover types to the drought-areas. Moreover, we will add a map to the supplementary, which depicts the CLC-classes used for the analyses across Europe. However, given the nature of the utilized data for land-cover classification, it is not possible to differentiate these classes further. We are aware that this is associated with potential problems, a fact that we recommend to take into consideration in follow up studies in lines 334 to 337 of the initial submission.*

*Also, we would like to emphasize the importance of Fig. 5 for the main aim of the paper, i.e. to compare 2003 with 2018. This figure fulfills two different tasks. On the one hand, we use it to show the absolute areas of drought affected ecosystems but separated by land-cover type. On the other hand, we use Fig. 5 to visualize how the quantiles are distributed within the 3 different classes of CWB-anomalies which indicates the effect of CWB on quantile-distributions. We are therefore convinced, that Fig. 5 provides an important message, namely that the area featuring extreme drought (CWB <-2) was several times higher in 2018 (already mentioned in the results of CWB anomalies) but that this larger area was unevenly distributed among the different vegetation types. That is, while the affected area of pastures was lower in 2018, arable land, as well as mixed and coniferous forests expressed much larger areas in comparison to 2003. We agree, that information is also needed on the relative contribution of each land-cover type to the drought affected areas, which is the reason why we also supplied Fig. S3 and report the corresponding results in lines 245-246 in the initial submission. We however deem these results as of lower importance, since they 'just' show that most (but not all) of the relative contributions were similar in 2003 and 2018, thus hide the fact that a much larger area was affected in 2018. Since the main emphasis of the paper is to compare the absolute drought effects, histograms relying on absolute areas are mandatory.*

**Statistical indices**
Finally, another aspect of the manuscript that makes it less convincing
is the choice of the figures and complex statistical indices derived. For example
figure 5 is hard to interprete. What is the implication of high NDVIs combined with low
CWB? In general, the methods section is very concise, making it easy to read but also
lacking some key points to help the reader understand the many indices used. For
example even though they are widely used NDVI and EVI should be defined. Also,
heat load variable is not defined and is sometimes written heat load and sometimes
heat-load. If it is simply Tmax call it this.

*Thanks for your suggestion to define NDVI and EVI, which we will do, in tandem with generally elaborating the methods section to better explain what we did and why we did this. Moreover, we will rename heat load to Tmax.*

*In fact, all data transformations are widely known, frequently used across environmental sciences, and rather simple: z-transformation to transform raw values into anomalies, ranking the data to obtain quantiles. The presented descriptive statistics are also not complex: means and standard deviations as well as total areas passing specific thresholds. As for inferential statistics, we present simple linear ordinary least squares regressions between NDVI-quantiles and CWB, and a mixed-effect model, of which the latter might be considered as more involved, but is developing towards a highly recommended standard for inference in ecological studies (e.g., Bolker et al. 2009).*

*Bolker, B., Brooks, M., Clark, C., Geange, S., Poulsen, J., Stevens, M., & White, J. (2009). Generalized linear mixed models: a practical guide for ecology and evolution. Trends in Ecology and Evolution. Retrieved from http://dx.doi.org/10.1016/j.tree.2008.10.008*

*As mentioned in the reply to your previous comment, Fig. 5 serves two tasks, i.e. 1) highlighting the much higher area of affected ecosystems with a different distribution among land-cover types and 2) exemplifying the effect of CWB on the distribution of quantiles. Under normal conditions, quantiles will express a uniform distribution, while a skewed distribution indicates abnormal conditions, such as for the classes with CWB < -2 and CWB > 2. Showing the skewed quantile distributions directly leads to the modelling exercises shown in Fig. 7.*
*The outlined amount of combinations with high NDVI but low CWB is relatively low and much lower than expected if assuming a uniform quantile-distribution due to the significant impact of CWB-deficit on NDVI quantiles. The reason for why we yet observe high NDVI-values despite low CWB is likely related to the different spatial resolutions of the used products, i.e. climate data with 0.5° resolution and satellite data with 231 m resolution. That is, within one climate grid cell elevational differences and variations in groundwater levels (e.g. along rivers or lake shores) will modify the vegetation response to gridded CWB, since the coarse resolution of the climate data neither capture small-scale climatic variations due to elevation nor represent groundwater levels. We will add a corresponding paragraph to the discussion to make this clearer.*

---

## Author Comment (AC2) · 30 Sep 2019

This is a well-written study comparing the European heatwaves of 2003 and 2018. Comparison of climatological data and vegetation indices lead to the conclusion that the 2018 heatwave was more severe than the 2003 heatwave. However, substantial regional differences occur. The idea behind this study is interesting, but the study misses out on several aspects needed to support the conclusions. Especially the lack of the temporal patterns in weather data makes it hard to evaluate the results. No time series for temperature, precipitation or drought indices are shown to illustrate the heatwave patterns of both years. Moreover, end of July was chosen as the study period for the impact on vegetation, hence ignoring any change that took place in August (e.g. the massive forest fires in Portugal early August 2018).

*Thank you for your constructive review. We agree with most of your comments and will modify our manuscript accordingly. Please find our detailed reply in the attached pdf-file.*

**Specific comments:**
L 96ff: I suggest to simply argue that you focused on March-Oct because that is the period of interest for vegetation dynamics and leaving out winter helps avoiding artefacts (e.g. snow cover, but also defoliation in deciduous systems).

*Thank you for this suggestion. We will modify the corresponding text accordingly.*

L 104-105: interpolation may create artefacts when searching for anomalies – especially when there are gaps during the drought episode under study. I suppose this is of minor importance for this study, because gaps are less likely during periods of drought (i.e., no clouds), but I wonder if the interpolation can be avoided. If not, the possibility for such artefacts should at least be discussed.

*As you pointed out correctly, the likelihood of gaps is very low in 2003 and 2018 since large parts of Europe were free of clouds during these events. However, in 2018 the Mediterranean experienced above average precipitation, which may have resulted in higher cloud coverage and thus missing values. To quantify the amount of missing values, we will compute maps for 2003 and 2018 which show the number of gaps that were filled in each of the two seasons and moreover add a paragraph to the discussion related to the influence of associated artefacts.*

L 119ff: NDVI and EVI are mainly greenness indicators. They may reflect photosynthesis, but not if photosynthesis changes without changing greenness. This is particularly relevant for drought. In this sense, EVI is better than NDVI (see Vicca et al 2016, Scientific Reports). I therefore advise to use the EVI results rather than the NDVI in this manuscript. It should also be clearly indicated what these VIs can reflect (and what not!). This is completely missing from the discussion of the current manuscript, but needs to be discussed (i.e., are we looking at green biomass/browning/defoliation. . . and what are the implications for e.g. legacy effects).

*In the revised version we will elaborate the corresponding methods section as proposed. We agree, that EVI has certain advantages over NDVI regarding its ability to detect specific changes in GPP as in Vicca et al., 2016. However, the study by Vicca et al., 2016 focused on GPP-reduction where no defoliation or leaf-coloration was observed. In contrast, both in 2003 and 2018 early leaf shedding and coloration was observed (see also below regarding additional phenological analyses undertaken). The question which VI to use for drought indices is widely*

*debated and depends on the purpose. For instance, Li et al. (2010 in Procedia of Environmental Sciences) found stronger correlations between NDVI and field observations in comparison to EVI. Moreover, NDVI is usually more chlorophyll sensitive and mainly reflects the photosynthetic activity (which we were mainly interested in) while EVI is more responsive to canopy structural variations. Also, in light of the findings by Vicca et al., 2016, it is hard to interpret the fact that the area with lowest quantiles was much larger for NDVI compared to EVI (Fig. 4 compared to Fig. S 4). That is, if a non-visible drought-response would have been missed by NDVI, the areas indicating a severe drought response should be larger for EVI, which is not the case. Finally, all of the cited studies which monitored drought-impacts by means of remote sensing made use of NDVI (Anyamba and Tucker, 2012; Bastos et al., 2017; Xu et al., 2011), wherefore concentrating on NDVI makes our work more comparable to previous studies. Since we from the beginning were aware of that the choice of VI matters, we included EVI in the supplementary to provide the full picture. To account for potential issues with regard to the chosen metric for quantifying drought we will add a paragraph to the discussion about NDVI-EVI comparison and mention the possibility to also assess other remotely sensed indices such as the photochemical reflectance index (Vicca et al., 2016) and solar induced fluorescence in future studies.*

L 141-142: awkward phrasing: Subsequently, we for 2003 and 2018 determined . . . should be: Subsequently, we determined the difference between 2003 and 2018 for the respective metric. . .

*Thank you for this suggestion. We will modify the corresponding text accordingly.*

L149ff: the timing of the heatwaves should be demonstrated with data to justify the choice to focus on end of July. Time series of temperature and CWB for e.g. France, Germany (which suffered from the heat in both 2003 and 2018), or even for the different regions (N, W, S, Central Europe). How sensitive is your analysis to the time choice? Are results similar if the analyses were repeated for end of August for example?

*We agree, that the timing of drought as well as the location of the drought epi-centers differs between 2003 and 2018. To better visualize the temporal development in 2018 and 2003 we will add supplementary material showing the temporal development of CWB and NDVI in the two years. Fig. 6 was particularly designed for this purpose, but we understand that we have to elaborate this part of the manuscript further. We will also perform the analyses depicted in Fig. 5 and 7 on the basis of a different selection of time slices for 2003 (DOY 241) and 2018 (DOY 209) to account for the differing timing of the droughts and discuss the results in light of the temporal development of the two events.*

L 150-151: VIs cannot be lower than 0. (anomalies can)

*Here, we disagree. Given the formulation of VIs in the numerator (NIR-RED)/(NIR+RED), VIs range from -1 to +1 and will become smaller than zero if the reflectance in the red spectrum is larger than in the NIR. Also, we want to point out that we were not using anomalies of VIs, since VIs have a bounded distribution (See also below). That is, instead we computed quantiles. All of this is described in detail in lines 150-156 of the initial submission.*

L 159: What was done with pixels where land cover changed between 2003 and 2018? Was that even considered? (I don't think it will have a big impact on the analyses,

but it's worth a mention).

*Thanks for this suggestion. We did not consider the changes in land-cover between 2003 and 2018 but will add a map to the supplementary to indicate which pixels of the considered CLC-classes changed their class from 2000 to 2018 and exclude all corresponding pixels from the analyses to avoid artefacts.*

L191: 0.55 should be 55% I suppose.

*Thanks for spotting the error. Will be changed.*

Fig.1: why was the timing April-July chosen for these figure? This is not motivated in the text I think a time series with weather data would be very helpful to evaluate this choice (see earlier comment). I noticed that this is briefly mentioned in the discussion (l. 320), but data are not shown. Please do show these data.

*We agree with the referee and we will improve our manuscript regarding this aspect. The end of the period was set to July since we were assessing end of July VI-quantiles. The beginning was set to April, since the integration over this period revealed the strongest relationship with end of July VI-quantiles. That is, adding or removing months before or after April lowered the relationship between CWB and VI-quantiles. As mentioned earlier, we will add several time-slices to our analyses as supplementary material and elaborate the analyses using CWB of different time-steps for modelling VI quantiles (e.g. if assessing NDVI at DOY 241 for 2003 consider the period May-Aug instead of Apr-July).*

Fig.4: It is unclear where VI-deviations from the mean were significant. Please clarify, also in the text.

*This seems to be a misunderstanding of the shown values. Since we used quantiles and not anomalies, it is not possible to derive p-values for the presented values. The reason for using quantiles instead of anomalies (bounded distribution of VIs) is given in lines 150-156.*

l.240ff: A map with vegetation types is missing to illustrate where the different vegetation types occur and how the differences in impact for the different vegetation types correspond with the regional differences (e.g. Scandinavia being dominated by conifer forests).

*In the map depicting which pixels changed their CLC between 2000 and 2018 (see above) we will also depict the representation of CLC-classes.*

Fig.6: consider moving to appendix, adding instead a figure with time series for weather data.

*We think that it is important to show the temporal development of NDVI-quantiles and will complement Figure 6, according to your suggestion, with time series for weather data.*

L 251: Fig. 7 shows EVI, not NDVI. The text is about NDVI. (I suggest to focus on EVI for in the main document and move NDVI to appendix – see earlier comment).

*Thank you for spotting an error in the axis labelling. Actually, Fig. 7 shows NDVI and not EVI (the same results for EVI are shown in the supplementary, Fig S7). Regarding your suggestion to focus on EVI, please see our detailed reply above.*

L328ff: Portugal suffered from severe wildfires in 2018. This is not included in the analyses because the fires occurred mostly in August and the analyses are only for April-July.
Other important events may be missed out because of the choice for April-July.

*As mentioned above, we will add supplementary information which will allow to see the impact of forest fires in NDVI and EVI. This will allow to spot other possibly important events. However, we believe that, given their comparably low relative spatial extent (please don't get us wrong - we are aware that these fires were massive but in relation to all European forests rather contribute a lower proportion of forest area,) it seems likely that the overall impact of these regional wildfires on our analyses is low.*

L 340ff: I suggest to include in this part of the discussion some text on the relationship between ecosystem types and climatic regions and how this may/may not influence the interpretation (see also earlier suggestion for figure addition).

*In complementation to Fig. 5, we will add a figure to the supplementary depicting the spatial contributions of the landcover types to the drought-affected areas. Moreover, we will elaborate the discussion in section 4.2.2 dealing with the drought response of ecosystem types across different climate zones.*

L358: public news references are not appropriate for this statement.

*We agree, but to date, we have not found any scientific publication that reports an earlier leaf shedding for broadleaved trees in 2018. To provide a more robust background to this statement, we will add results from a new analysis of DWD-phenological data to the supplementary which clearly highlights the earlier leaf-shedding of beech in summer 2018 in Germany.*

---

## Author Response (AR1)

**Letter of reply and track-changes manuscript for BG-2019-286.**

Dear Prof. Luyssaert, Dear reviewers,

Thank you very much for considering our manuscript for publication and the constructive feedback on our initial submission. We have now thoroughly revised our manuscript 'Quantifying impacts of the drought 2018 on European ecosystems in comparison to 2003' in light of the interactive discussion'. Major changes undertaken in course of the revision are:

- Adding spatiotemporal analyses of climate and remote sensing data (including four supplementary videos), to depict the differing temporal development of the two drought events

   - Changed the comparison between 2003 and 2018 to now emphasize on the peak of drought in both years, i.e. August 2003 vs. July 2018.

   - Elaborated the methods and discussion sections to clarify potential ambiguity and strengthen the discussion on the spatiotemporal differences of drought impacts.

As can be seen in the revised document, the general message of the study – which was in parts challenged by the reviewers – remains consistent, namely that the drought 2018 superseded the one of 2003 both with regard to climatic features but also ecosystem response. The added analyses provide more insight into the development of the two differing droughts. Refined discussions on the differences and hypotheses for their explanation now more clearly stimulate research questions for future studies. We are therefore confident, that our revision and reply addresses all of the points raised by the reviewers which helped us to significantly improve our manuscript, even if we did not incorporate all of the reviewer's suggestions for reasons that are explained in our point-by-point reply below. We therefore hope that these amendments justify publication in *Biogeosciences.*

Please find our point by point reply (your comments in normal font, ours in italics with text citations in bold) as well as the track-changes document of our revision below.

**Anonymous Referee #1**

In this manuscript, the authors compare the climatological features of two intense drought events over Europe, the 2003 and 2018 heatwaves. From a climatological analysis, they carry on to analyze the effect of both events on European vegetation. The authors' results are based on a suite of statistical analyses of MODIS-based vegetation indices combined with a widely used land cover map. Their main conclusion is that the 2018 heatwave had a stronger effect on European vegetation than the 2003 one.

**General comments**

Overall the manuscript is well written and follows a logic questioning line going from the climatology of the heatwave to the effect on vegetation. However, I am not convinced by the relevancy of the way the main question is addressed. From the first climatological data shown by the authors it is obvious that the 2003 and 2018 events are very different in terms of location (baltic countries in 2018 versus central Europe in 2003) and timing (july was the beginning of the 2003 heatwave whereas it was the end of the 2018 one) and, even though it is not shown, in initial conditions. These crucial differences are however mostly ignored in the way the analyses are designed, potentially pointing to severe flaws in the results, that I detail below. This observation leads me to suggest more detailed analyses be carried out before publication.

*Thank you very much for your constructive review which highlights important aspects and potential ambiguities that made us improve our analyses and consequently the manuscript. Major changes to the manuscript in light of your comments are:*

1) *Visualizing the spatiotemporal development of climatic conditions using time-series and animated videos (Fig. S5 and supplementary videos V1 and V2* http://doi.org/10.5446/44027 *and* http://doi.org/10.5446/44028*) to better quantify the climatic conditions of the two events,*

2) *Provide further supplementary videos (V3 and V4* http://doi.org/10.5446/44029 *and* http://doi.org/10.5446/44030*) which visualize the temporal development of NDVI and EVI quantiles using maps and histograms (as in Fig. 4) from beginning of May until end of October to allow for assessing the temporal development of ecosystem response to the two drought events,*

3) *Comparing and modelling the ecosystem response representative of the peak of drought in each year (i.e. DOY 209 in 2018 vs. DOY 241 in 2003) instead of for DOY 209,*

4) *Elaborating the discussion regarding the outlined spatiotemporal effects.*

*All of these additional analyses confirm the initial impression that the drought 2018 was more severe compared to 2003. However, we now pay more attention to outline the different spatial impact and timing of the drought 2018 compared to 2003 which is related to the more northward location of the drought center. Please find our more detailed responses below.*

**Detailed comments**

**Effect of heatwave timing, duration and legacy**

The analyses carried out, only encompassing the greenness index in july of both years, seem too superficial to draw the far-reaching conclusions the authors make. First on heatwave timing, the stronger NDVI signal in july 2018 compared to july 2003 could be the result of the difference in timing. This point is ignored when concluding that the 2018 heatwave had a stronger effect on European ecosystems than the 2003 one.

*We agree that the different timing of the drought and heatwaves likely resulted in a different timing of vegetation response. To better visualize the observed temporal development, we added supplementary videos which depict the temporal development of NDVI and EVI from beginning of May until end of October (supplementary videos V3 and V4). Moreover, we added time-series comparable to those shown in Fig. 6 for TMAX and CWB to provide a direct comparison between the climatic features of the drought affected areas in 2003 and 2018 (Fig. S5) as well as videos depicting the course of TMAX and CWB from January through October for both years (V1 and V2). These results further strengthen our initial conclusion that the 2018 event overall had a larger extent and a stronger effect on European ecosystems compared to 2003. In our revision we have pointed out more clearly the differences between the timing and location of drought, i.e. we added several paragraphs to the discussion about the additional analyses and the importance of drought timing and location, for example in*

*Line 347: Moreover, the two drought events differed regarding their timing and location, with 2018 featuring an earlier peak of drought (July vs. August in 2003) and a more northward centre around the Baltic Sea vs. the Mediterranean in 2003.*

*Line 425: The earlier timing possibly also triggered a stronger response to the drought in 2018, particularly in Northern Europe where it began as early as May, i.e. at the beginning of the growing season (Fig. S5 and V2). Northern European forests are dominated by coniferous forests that to a large degree consist of Norway spruce and Scots pine, i.e. two tree species that have been frequently reported to suffer under drought (e.g. Buras et al., 2018; Kohler et al., 2010; Rehschuh et al., 2017; Rigling et al., 2013). Moreover, coniferous forests made up a high share of the drought affected ecosystems in 2018 (Fig. S6). In combination, the potentially stronger reaction of high latitude coniferous forests may partly explain the observed stronger coupling between CWB anomalies and VI quantiles in 2018 compared to 2003.*

Second, as mentioned as a discussion item, the legacy of water balance can be very important for heatwave effects on forests especially. Even though this is be a major point underlying the relevancy of the question asked, no data or analysis shown tries to compare the water conditions prior to both heatwaves. An analysis of both heatwaves time evolution and the comparison of each heatwave's end month, that could then be different from one to the other could be an option.

*We agree that the water balance legacy effects are relevant, wherefore we added analyses to the manuscript that address this issue. In particular, we visualized the development of integrated CWB over time, i.e. including preceding winter conditions and extending until the end of the growing season (Fig.*

*S5) and added corresponding supplementary videos (V1 and V2). We acknowledge your suggestion to compare different months with each other. We now compare VI quantiles representing DOY 241 in 2003 and DOY 209 in 2018 in Figs. 4 and 5, and use the corresponding VI-quantiles and CWB for the modelling exercise presented in Fig. 7 and the mixed model. Having added these analyses to the manuscript now*
*provides a more comprehensive picture of the difference between the two drought events.*

**Heatwave location defining the ecosystems affected**
Another point that undermines the results presented is the comparison of vegetation types in absolute
terms without a prior description of vegetation types affected in each case that might be very different. Even though this information is essential to make sense of a comparison of the effects on vegetation of two climatic events, it is hardly discussed and made very hard to see in the way the data are processed. For example the varying y-axis ranges in figure
hide the relative weights of each vegetation type with water deficit or water surplus.
Figure S3 might be more explicit to this regard by considering relative vegetation cover.

*Maybe we misunderstood your comment, but we struggle to reproduce your statement that the effects of the different locations of the drought are 'hardly discussed'. Section 4.2.2 particularly deals with the possible effect of the differing drought epicenters, where we highlight that the different ecosystems'*
*adaptation to drought likely caused the stronger ecosystem response in 2018. To account for your critique, we have added Fig. S6 which depicts the share of ecosystems that were affected by the two drought events, respectively. This figure further highlights the fact that the drought-affected areas differ between 2003 and 2018, which leads to varying absolute areas as well as different vegetation types that were affected. However, given the nature of the data available for land-cover classification, it is not*
*possible to differentiate these classes further. We are aware that this is associated with potential problems, a fact that we recommend to take into consideration in follow-up studies. Moreover, we now emphasize the potentially stronger response of high latitude coniferous forests in comparison to Mediterranean forests. Please see the two text excerpts from chapter 4.2.2 below:*

***Line 426: Northern European forests are dominated by coniferous forests that to a large degree consist of Norway spruce and Scots pine, i.e. two tree species that have been frequently reported to suffer from drought (e.g. Buras et al., 2018; Kohler et al., 2010; Rehschuh et al., 2017; Rigling et al., 2013). Moreover, coniferous forests made up a high share of the drought affected ecosystems in 2018 (Fig. S6). In combination, the potentially stronger reaction of high latitude coniferous forests may partly explain the observed stronger coupling between CWB***
***anomalies and VI quantiles in 2018 compared to 2003.***
***Line 439: Nevertheless, our hypotheses explaining the stronger coupling of VI quantiles with CWB in 2018 need further support, e.g. by studying the sensitivity and coupling of plant productivity with climatic properties for the considered land-cover classes as for instance done by (Anderegg et al., 2018). A sub-classification of land-cover classes seems to be reasonable for such an analysis, in order to account for possibly differing drought-***

*sensitivities of ecosystems represented by one specific land-cover class. For instance, Mediterranean coniferous forests comprise different species and are thus likely better adapted to drought than boreal coniferous forests in Scandinavia.*

*Also, we would like to emphasize the importance of Fig. 5 for the main aim of the paper, i.e. to compare 2003 with 2018. This figure fulfills two different tasks. On the one hand, we use it to show the absolute areas of drought affected ecosystems but separated by land-cover type. On the other hand, we use Fig. 5 to visualize how the quantiles are distributed within the 3 different classes of CWB-anomalies which indicates the effect of CWB on quantile-distributions. We are therefore convinced, that Fig. 5 provides an important message, namely that the area featuring extreme drought (CWB <-2) was larger in 2018 (already mentioned in the results of CWB anomalies but now also shown in Fig. S6) but that this higher area was unevenly distributed among the different vegetation types. For instance, while coniferous and mixed forests reveal a much larger spatial area featuring lowest VI-quantiles in 2018 compared to 2003, broadleaved forests revealed opposite patterns, i.e. a larger area of lowest quantiles in 2003.*
*We agree, that information is also needed on the relative contribution of each land-cover type to the drought affected areas, which is the reason why we also supplied Fig. S3 (now Fig. S8) and report the corresponding results in the results section. However, since the main emphasis of the paper is to compare the absolute drought effects, histograms relying on absolute areas are mandatory.*

**Statistical indices**
Finally, another aspect of the manuscript that makes it less convincing
is the choice of the figures and complex statistical indices derived. For example
figure 5 is hard to interprete. What is the implication of high NDVIs combined with low
CWB? In general, the methods section is very concise, making it easy to read but also
lacking some key points to help the reader understand the many indices used. For
example even though they are widely used NDVI and EVI should be defined. Also,
heat load variable is not defined and is sometimes written heat load and sometimes
heat-load. If it is simply Tmax call it this.

*We agree that the methods should be sufficiently precise to enable reproducing the analyses, and have added mathematical definitions of NDVI and EVI and generally elaborated the methods section to better explain what we did and why we did this. According to the reviewers' suggestion, we renamed heat load to Tmax.*

*However, we disagree with your statement regarding the application of complex statistical indices. In fact, many of the applied statistics are rather simple, i.e. transformation of processed values into anomalies (a simple z-transformation) and quantiles (a simple ranking) and then providing mostly descriptive statistics. Moreover, we show simple ordinary linear regressions between NDVI-quantiles*

*and CWB. The only more complex statistical approach is related to the mixed-effects model, which however, after being introduced in 1918, is becoming more and more common in environmental sciences.*

*As mentioned in the reply to your previous comment, Fig. 5 serves two tasks, i.e. 1) highlighting the much* higher area of affected ecosystems with a different distribution among land-cover types and 2) *exemplifying the effect of CWB on the distribution of quantiles. Under normal conditions, quantiles will express a uniform distribution, while a skewed distribution indicates abnormal conditions, such as for the classes with CWB < -2 and CWB > 0. Showing the skewed quantile distributions directly leads to the modelling exercises shown in Fig. 7.*

*The outlined amount of combinations with high NDVI but low CWB is relatively low and much lower than expected if assuming a uniform quantile-distribution due to the significant impact of CWB-deficit on NDVI quantiles. The reason for why we yet observe high NDVI-values despite low CWB is likely related to the different spatial resolutions of the used products, i.e. climate data with 0.5° resolution and satellite* data with 231 m resolution. That is, within one climate grid cell elevational differences and variations in *groundwater levels (e.g. along rivers or lake shores) may modify the vegetation response to gridded CWB, since the coarse resolution of the climate data neither capture small-scale climatic variations due to elevation nor represent groundwater levels. We have added two corresponding paragraphs to chapter 4.2.1 of the discussion to make this clearer:*

*Line 384: We want to note that we also observed small areas (ca. 1000 km² for each of the considered land cover classes) of high quantiles in regions featuring extreme drought. This observation is likely related to the different resolution of climate data (0.5°, thus ca. 50 km x 50 km) vs. MODIS (231 m x 231 m). Using such relatively coarse climate data for quantifying drought neglects elevation-driven climatic variations within grid cells (Zang* et al., 2019) in contrast to the relatively higher resolution of MODIS which likely captures such differences. For *instance, mountain ranges – which on average feature cooler temperatures (thus less evapotranspiration) and higher precipitation – likely feature less extreme water deficit compared to surrounding lowlands which may cause higher VI values than one would expect if only considering the drought classification of the corresponding climate grid cell. In addition, groundwater availability – which may vary within climate grid cells, particularly* in proximity to rivers and lakes – may locally modulate plant water availability, possibly explaining some of the *observations of high VI quantiles under extreme drought.*

*Line 394: Furthermore, quantifying drought on anomalies alone bears the risk to erroneously classify regions that actually feature water surplus (CWB > 0) as being drought-affected (Zang et al., 2019). This is likely to happen in regions which usually have high water surplus (e.g. the Norwegian west-coast) and thus may feature* more than two negative CWB standard deviations even though raw CWB is positive (Zang et al., 2019). For both *2003 and 2018, 5 (1) percent of climate grid cells featured positive CWB even though CWB anomalies indicated*

*(extreme) water deficit (CWB < 0 and CWB < -2 SD, see also Fig. S13). This false classification may explain some of the observed highest quantiles in regions that were classified as featuring extreme water deficit. Since we currently lack a more advanced drought-metric that compensates for such effects, we followed the commonly applied approach and used standardized CWB for our analyses. To show the full picture, we at the same identify potential biases caused by this approach (Fig. S13) as proposed by Zang et al. (2019). Given the relatively low number of 'false' extreme drought classifications (1 percent for each year) we estimate the impact on our analyses to be generally low.*

**Anonymous Referee #2**

This is a well-written study comparing the European heatwaves of 2003 and 2018. Comparison of climatological data and vegetation indices lead to the conclusion that the 2018 heatwave was more severe than the 2003 heatwave. However, substantial regional differences occur. The idea behind this study is interesting, but the study misses out on several aspects needed to support the conclusions. Especially the lack of the temporal patterns in weather data makes it hard to evaluate the results. No time series for temperature, precipitation or drought indices are shown to illustrate the heatwave patterns of both years. Moreover, end of July was chosen as the study period for the impact on vegetation, hence ignoring any change that took place in August (e.g. the massive forest fires in Portugal early August 2018).

*Thank you for your constructive review. We agree with most of your comments and have modified our manuscript accordingly. Please find our detailed responses below.*

**Specific comments:**

L 96ff: I suggest to simply argue that you focused on March-Oct because that is the period of interest for vegetation dynamics and leaving out winter helps avoiding artefacts (e.g. snow cover, but also defoliation in deciduous systems).

*Thank you for this suggestion. We have modified the corresponding text accordingly:*

***However, since we were only interested in VI time series during the growing season, we only considered the period from beginning of March (DOY 64) to end of October (DOY 304) for the definition of valid pixels.***

L 104-105: interpolation may create artefacts when searching for anomalies – especially when there are gaps during the drought episode under study. I suppose this is of minor importance for this study, because gaps are less likely during periods of drought (i.e., no clouds), but I wonder if the interpolation can be avoided. If not, the possibility for such artefacts should at least be discussed.

*As you pointed out correctly, the likelihood of gaps is very low in 2003 and 2018 since large parts of Europe were free of clouds during these events. However, in 2018 the Mediterranean experienced above average precipitation, which may have resulted in higher cloud cover and thus missing values. To quantify the amount of missing values, we computed maps for 2003 and 2018 which show the number of gaps that were filled in each of the two seasons and added a corresponding paragraph to the methods section in which we state that given a low number of gaps for the majority of pixels the influence on our analyses is likely low:*

*Line 108: To visualize the potential influence of this gap-filling procedure, we provide Fig. S1 which depicts the number of gaps filled in 2003 and 2018. Since 2/3 of pixels rendered one or zero gaps, and 95 % rendered five or less gaps (i.e. less than one third of corresponding images missing, thus presumably sufficient data for meaningful interpolation) we assume any potential biases caused by the gap-filling procedure to be marginal.*

L 119ff: NDVI and EVI are mainly greenness indicators. They may reflect photosynthesis, but not if photosynthesis changes without changing greenness. This is particularly relevant for drought. In this sense, EVI is better than NDVI (see Vicca et al 2016, Scientific Reports). I therefore advise to use the EVI results rather than the NDVI in this manuscript. It should also be clearly indicated what these VIs can reflect (and what not!). This is completely missing from the discussion of the current manuscript, but needs to be discussed (i.e., are we looking at green biomass/browning/defoliation. . . and what are the implications for e.g. legacy effects).

*We agree, that EVI has certain advantages over NDVI regarding its ability to detect specific changes in GPP as in Vicca et al., 2016. However, the study by Vicca et al., 2016 focused on GPP-reduction where no defoliation or leaf-coloration was observed. In contrast, both in 2003 and 2018 early leaf shedding and coloration was observed (see also Fig. S14 regarding phenological analyses undertaken). The question which VI to use for drought indices is widely debated and depends on the purpose. For instance, Li et al. 2010 (in Procedia of Environmental Sciences) found stronger correlations between NDVI and field observations in comparison to EVI. Moreover, NDVI is usually more chlorophyll sensitive and mainly reflects the photosynthetic activity (which we were mainly interested in) while EVI is more responsive to canopy structural variations (see e.g. Huete et al., 2002). Also, in light of the findings by Vicca et al., 2016, it is hard to interpret the fact that the area with lowest quantiles was larger for NDVI compared to EVI (Fig. 4 compared to Fig. S 4). That is, if a non-visible drought-response would have been missed by NDVI, the areas indicating a severe drought response should be larger for EVI, which does not seem to be the case. Finally, all of the cited studies which monitored drought-impacts by means of remote sensing made use of NDVI (Anyamba and Tucker, 2012; Orth et al., 2016; Xu et al., 2011), wherefore focusing on NDVI makes our work more comparable against previous studies. Since we were aware from the beginning that the choice of VI matters, we included EVI in the supplementary to provide the full picture. We have elaborated the corresponding methods section as proposed and added a paragraph to the discussion about NDVI-EVI comparison and outline the possibility to also assess other remotely sensed indices such as the photochemical reflectance index and solar-induced fluorescence in future studies. We have added the following paragraphs in*

*Line 126: Both NDVI and EVI are considered as proxy for photosynthetic carbon fixation, and thus allow for assessing possible changes in productivity in dependence of environmental conditions (Huete et al., 2006; Myneni et al., 1995; Xu et al., 2011). NDVI has earlier been used in the context of drought monitoring (Anyamba and Tucker, 2012) and assessing impacts of drought on ecosystems on large scales (Orth et al., 2016; Xu et al., 2011). While NDVI relies on information derived from the red and near infrared spectra (see equation 1) EVI*

*additionally makes use of the blue spectrum (see equation 2) to reduce atmospheric disturbance and influence of the understory:*

$$NDVI = \frac{NIR - RED}{NIR + RED} \quad (1)$$

$$EVI = G \cdot \frac{NIR - RED}{NIR + C_1 \cdot RED - C_2 \cdot BLUE + L} \quad (2)$$

*With G being the gain factor, $C_1$ and $C_2$ being the spectrum-specific coefficients of the aerosol resistance term, and L the canopy background adjustment term ($G = 2.5$, $C_1 = 6$, $C_2 = 7.5$, $L = 1$ for MODIS EVI, see Huete et al., 2002). Given these definitions, NDVI is more chlorophyll sensitive, while EVI is more sensitive to canopy structural variations (Huete et al., 2002). Thus, NDVI is more likely to reflect changes in leaf coloration as for instance in course of premature leaf senescence under drought, whereas EVI may better reflect early leaf shedding. For reasons of simplicity and to render our results comparable to previous studies which all used NDVI, we focus on results derived from NDVI. To provide the full picture, results derived from EVI are shown in the supplementary information which generally confirm the results based on NDVI.*

*Line 405: Finally, we want to stress that the choice of VI used for quantification of the ecosystem response to drought matters. While Li et al. (2010) found stronger relationships and lower errors between NDVI and ground observations made in grasslands, shrublands and forest in comparison to EVI, Vicca et al. (2016) reported EVI to be more sensitive to reductions in GPP that were not reflected by leaf coloration or early leaf senescence. Given the ongoing discussion, we present results from both VIs which generally support each other (e.g. comparison between Figs. 4 and S7, Figs. 7 and S12) but focused on NDVI to make our contribution directly comparable to previous studies dealing with drought impacts on ecosystems (Anyamba and Tucker, 2012; Orth et al., 2016; Xu et al., 2011). To complement our VI-based assessment, future studies may consider the use of solar-induced fluorescence which is known to be more sensitive to terrestrial photosynthesis (Li et al., 2018) and/or the photochemical reflectance index (Vicca et al., 2016).*

L 141-142: awkward phrasing: Subsequently, we for 2003 and 2018 determined
. . . should be: Subsequently, we determined the difference between 2003 and 2018 for
the respective metric. . .

*Thank you for this suggestion. We have modified the corresponding text accordingly.*

L149ff: the timing of the heatwaves should be demonstrated with data to justify the choice to focus on end of July. Time series of temperature and CWB for e.g. France, Germany (which suffered from the heat in both 2003 and 2018), or even for the different regions (N, W, S, Central Europe). How sensitive is your analysis to the time choice? Are results similar if the analyses were repeated for end of August for example?

*We agree, that the timing of drought as well as the location of the drought differs between 2003 and 2018. To better visualize the temporal development in 2018 and 2003 we have added supplementary videos V1-V4 (http://doi.org/10.5446/44027, http://doi.org/10.5446/44028, http://doi.org/10.5446/44029, http://doi.org/10.5446/44030) and Figs. S5 and S11 depicting the temporal development of TMAX, CWB, NDVI and EVI in the two years. Fig. 6 was particularly designed for this purpose, but we understand that*
*we had to elaborate this part. We now also perform the analyses depicted in Fig. 5 and 7 on the basis of a different selection of time slices for 2003 (DOY 241) and 2018 (DOY 209) to account for the differing timing of the droughts and have elaborated the discussion about potential impacts of the differing spatiotemporal development of the two drought events:*

***Line 348: Moreover, the two drought events differed regarding their timing and location, with 2018 featuring an earlier peak of drought (July vs. August in 2003) and a more northward centre around the Baltic Sea vs. the Mediterranean in 2003.***

***Line 426: The earlier timing possibly also triggered a stronger response to the drought in 2018, particularly in Northern Europe where it began as early as May, i.e. at the beginning of the growing season (Fig. S5 and V2).***
***Northern European forests are dominated by coniferous forests that to a large degree consist of Norway spruce and Scots pine, i.e. two tree species that have been frequently reported to suffer from drought (e.g. Buras et al., 2018; Kohler et al., 2010; Rehschuh et al., 2017; Rigling et al., 2013). Moreover, coniferous forests made up a high share of the drought affected ecosystems in 2018 (Fig. S6). In combination, the potentially stronger reaction of high latitude coniferous forests may partly explain the observed stronger coupling between CWB***
***anomalies and VI quantiles in 2018 compared to 2003.***
L 150-151: VIs cannot be lower than 0. (anomalies can)

*Here, we disagree. Given the formulation of VIs in the numerator (NIR-RED)/(NIR+RED), VIs range from -1 to +1 and will become zero if the reflectance in the red spectrum is larger than in the NIR. Also,*
*we want to point out that we were not using anomalies of VIs, since VIs have a bounded distribution. That is, instead we computed quantiles. All of this is described in section 2.2:*

***Line 177: To quantify the response of European ecosystems to the two drought events, we focused on end-of-August (DOY 241) and end-of-July (DOY 209) VI values for 2003 and 2018, respectively. The selection of these***
***particular dates was based on the peak of climatological drought (see previous paragraph). Since VI features a bounded distribution (values between -1 and +1), we could not apply a standardization approach as for the climate variables. Therefore, we computed for each VI time series its end-of-July quantiles over the 19 years similar to Orth et al. (2016).***

L 159: What was done with pixels where land cover changed between 2003 and 2018?
Was that even considered? (I don't think it will have a big impact on the analyses,
but it's worth a mention).

*Thanks for this mention. We have updated our analyses to now only consider pixels which featured*
*consistent land cover in 2000 and 2018. We have added supplementary Fig. S3 which depicts which pixels were used.*

L191: 0.55 should be 55% I suppose.

*Thanks for spotting the error. We have corrected it accordingly.*

Fig.1: why was the timing April-July chosen for these figure? This is not motivated in the text I think a time series with weather data would be very helpful to evaluate this choice (see earlier comment). I noticed that this is briefly mentioned in the discussion (l. 320), but data
are not shown. Please do show these data.

*We have added supplementary videos V1 and V2 and Fig. S5 to our analyses which depict the spatiotemporal development of TMAX and CWB. Moreover, we now compared the VI-quantiles and corresponding integrated climate parameters for the peak of drought in each year, i.e. August 2003 vs.*
*July 2018.*

Fig.4: It is unclear where VI-deviations from the mean were significant. Please clarify, also in the text.

*This seems to be a misunderstanding of the shown values. Since we used quantiles and not anomalies, it*
*is not possible to derive p-values for the presented values. The reason for using quantiles instead of anomalies is mentioned in section 2.2 (see also reply to your previous comment).*

l.240ff: A map with vegetation types is missing to illustrate where the different vegetation types occur and how the differences in impact for the different vegetation types correspond with the regional
differences (e.g. Scandinavia being dominated by conifer forests).

*Fig. S3 has been added for this purpose. It depicts which pixels were eventually used and also to which land-cover class they belong.*
Fig.6: consider moving to appendix, adding instead a figure with time series for weather data.

*We have kept Fig. 6 in the main part of the manuscript but have complemented it with Fig. S5 which depicts the temporal development of TMAX and CWB for Northern, Central, and Southern Europe.*

L 251: Fig. 7 shows EVI, not NDVI. The text is about NDVI. (I suggest to focus on EVI
for in the main document and move NDVI to appendix – see earlier comment).

*Thank you for spotting an error in the axis labelling. Actually, Fig. 7 shows NDVI and not EVI (the same results for EVI are shown in the supplementary, Fig S12). Regarding your suggestion to focus on EVI, please see our detailed reply above.*

L328ff: Portugal suffered from severe wildfires in 2018. This is not included in the analyses because the fires occurred mostly in August and the analyses are only for April-July. Other important events may be missed out because of the choice for April-July.

*We have added supplementary videos which will allow for assessing the temporal development of NDVI and EVI and consequently to spot other possibly important events. However, given the large extent of the European map, the wildfires are hardly seen in these videos, i.e. a few pixels in Portugal turn red in August but the majority of pixels remains blue. Moreover, we believe that given their comparably low relative spatial extent (please don't get us wrong - we are aware that these fires were massive but in*
*relation to all European forests rather contribute a lower proportion of forest area) it seems likely that the overall impact of these regional wildfires on our analyses is low. Since our main aim is to compare the overall impact of drought between 2003 and 2018, we would prefer to refrain from detailed mentions of single events. For instance, Sweden, Denmark, and Germany also suffered from massive forest fires in July 2018.*

L 340ff: I suggest to include in this part of the discussion some text on the relationship between ecosystem types and climatic regions and how this may/may not influence the interpretation (see also earlier suggestion for figure addition).

*In complementation to Fig. 5, we have added figure S6 to the supplementary depicting the spatial contributions of the landcover types to the drought-affected areas. In section 4.2.2 we elaborated the discussion how the different adaptation of affected ecosystems probably affects our results and interpretation, i.e. the observed stronger ecosystem response of 2018 is probably related to the fact that less adapted ecosystems were hit by the drought:*

**Line 427: Northern European forests are dominated by coniferous forests that to a large degree consist of Norway spruce and Scots pine, i.e. two tree species that have been frequently reported to suffer from drought (e.g. Buras et al., 2018; Kohler et al., 2010; Rehschuh et al., 2017; Rigling et al., 2013). Moreover, coniferous forests made up a high share of the drought affected ecosystems in 2018 (Fig. S6). In combination, the potentially stronger**
**reaction of high latitude coniferous forests may partly explain the observed stronger coupling between CWB anomalies and VI quantiles in 2018 compared to 2003.**

L358: public news references are not appropriate for this statement.

*We agree and have added supplementary Fig. S. 14 which depicts the earlier timing of leaf senescence based on phenological data for Germany.*

[revised manuscript text omitted]

---

## Referee Report (RR1)

This study compares climatological data and vegetation indices between the 2003 and 2018 drought. It concludes that the 2018 drought response was more extreme i.a. due to a difference in the location and timing of the drought. The manuscript is well written and structured. I think the concerns of the reviewers in particular about the temporal and spatial differences between the two events have been adequately addressed.

I only have a few very minor comments:

Line 159: Shapiro-Wilk?

Line 225: 0.61% should be 61% I guess?

Line 495: '...we observed a different sensitivity of ecosystems to CWB between the two events...' In light of the discussion in 4.2.2, I think this should be rephrased. It is not necessarily the sensitivity of the ecosystems that is different but rather the location and timing of the events subjecting different ecosystems to drought, i.e. the 2018 event hit more sensitive ecosystems earlier in the growing season.

---

## Author Response (AR2)

Dear Prof. Luyssaert,

Thank you very much for considering our manuscript for publication in Biogeosciences. We have now carefully revised our manuscript in light of the reviews. In our revision we have:

1) Added additional analyses to the supplementary which only compare regions between 2003 and 2018 that were affected by extreme or moderate drought in both years (Figs. S11-13 as well as additional corresponding statements in methods, results, discussion and conclusions)
2) Included several statements as requested by the reviewers.

Please find a detailed point by point reply to the reviewers' comments below.

Altogether, we incorporated all of the amendments requested by the reviewers and hope that our manuscript now meets the requirements for publication in Biogeosciences.

With my best regard on behalf of all authors

Allan Buras

**Report #2**

This study compares climatological data and vegetation indices between the 2003 and 2018 drought. It concludes that the 2018 drought response was more extreme i.a. due to a difference in the location and timing of the drought. The manuscript is well written and structured. I think the concerns of the reviewers in particular about the temporal and spatial differences between the two events have been adequately addressed.

*Reply: Thank you very much for the positive evaluation of our study. We highly appreciate the time you have spent to review our paper.*

I only have a few very minor comments:

Line 159: Shapiro-Wilk?

*Reply: Yes, we have rephrased to Shapiro-Wilk.*

Line 225: 0.61% should be 61% I guess?

*Reply: Thank you for spotting this error. We have corrected to 61 %.*

Line 495: '...we observed a different sensitivity of ecosystems to CWB between the two events...' In light of the discussion in 4.2.2, I think this should be rephrased. It is not necessarily the sensitivity of the ecosystems that is different but rather the location and timing of the events subjecting different ecosystems to drought, i.e. the 2018 event hit more sensitive ecosystems earlier in the growing season.

*Reply: Here, we use the term sensitivity as in Anderegg et al. (2018) "Hydraulic diversity of forests regulates ecosystem resilience during drought" (Nature), where regression slopes are interpreted as sensitivity (and r² as coupling). To account for your comment, we have added the following statement to the corresponding sentence:*

*More specifically, we observed a different sensitivity of ecosystems to CWB between the two events and a differing sensitivity of land cover classes to drought, with pastures and agricultural fields expressing a higher sensitivity in comparison to forests, which probably was caused by the differing spatial extent of the two events thereby affecting presumably less drought-resistant ecosystems in 2018.*

**Report #3**

The authors have done a good job revising the manuscript. Below I list some (minor) comments and suggestions to further improve the manuscript and clarify a few points.

*Reply: Thank you very much for the positive evaluation of our manuscript. We greatly appreciate the time you have spent to review our study.*

Abstract:
Abbreviations not explained (NDVI, EVI, VI). Especially VI may need clarification for readers who are not familiar with remote sensing terminology.

*Reply: We now also explain the abbreviations in the abstract (L 15).*

Introduction:
l. 53: this sentence may need rephrasing. I suggest writing "… heatwave of 2003 was long considered the most extreme compound event… " because it doesn't come as a big surprise that 2018 was worse (and perhaps so was 2019?)

*Reply: We have amended the sentence accordingly.*

Materials and Methods:

l. 84-86: not totally clear. Did you use April-July for 2003 and May-August for 2018? Please clarify and motivate this choice better. Why not the same period (e.g. April-August) for both years?

*Reply: The selection of these two periods was chosen on the basis of climate data, which indicated the peak of the 2003 drought to occur in August, while in 2018 it occurred in July. In the first round of reviews we were asked to use those two periods to have a better comparison between the corresponding peaks of the two drought events. We have now added the explanation for this selection in this section and also refer to section 2.2 where this motivation is explained in detail.*

**These particular, differing periods were chosen, since they each represent the peak of drought for the corresponding year (see also section 2.2).**

l. 198: I suppose this should be Fig. S6 instead of Fig. S5 (which does not show land cover types)? Please also clarify the abbreviations used in Fig. S6 (and other supplementary figures).

*Reply: Thank you for spotting this error. We now refer to S6 and have clarified the abbreviations in the supplementary figures.*

Results:

Fig. 1: Legend mentions April-July. Should this be August (2003) and July (2018)? Or May-August and April-July? Similar remark for Fig. 3 (while in Fig. 2, the legend has been revised).

*Reply: Thank you for spotting this error. We have corrected the figure captions.*

Fig. 5: these results are of course strongly determined by the spatial differences in the heatwave of 2018 vs 2003. I would be interested to see the NDVI analysis for Central Europe only, where the heatwave was similarly extreme in both years. In other words, excluding N and S Europe, where the weather was very different for both years. This could give some extra substance to section 4.2.2.

*Reply: Thank you for this suggestion. We have now added supplementary Figs. S11-S13 which show similar histograms as in Fig. 5 but now only for the intersection of areas featuring extreme drought (CWB anomaly < -2) in both years as well as the intersection of areas featuring moderately dry conditions (-2 < CWB < 0). Since there was hardly any overlap between pixels featuring positive CWB anomalies, we refrained from depicting these comparisons, too. The resulting histograms underline the previously acquired findings that the drought of 2018 superseded the one of 2003. In particular, forest ecosystems featured a higher share of lower NDVI (and EVI for moderate drought areas) quantiles, which we interpret as a consequence of legacies from drought events before 2015. We have added corresponding statements in the methods, results, discussion, and conclusions:*

**Methods, section 2.2.:**

**Due to the different spatial patterns in drought severity between the two years, we determined the intersection area where the same CWB anomaly classes were observed in both 2003 and 2018. For this intersection area, we repeated the comparison of the 19 different quantiles between 2003 and 2018 for the five different land-cover by comparing the same pixels for each combination of land-cover class and CWB anomaly class between the two years. This was done, to avoid artefacts related to the fact that CWB anomaly classes were represented by different ecosystems in 2003 compared to 2018. Since the overlap for positive CWB anomalies was very low (altogether only 455 MODIS pixels, and thus no observation for some of the quantiles in some of the land cover classes), we refrained from computing those given their low representativity. Consequently, we only depict the comparison for extreme (CWB < -2) and moderate (-2 < CWB < 0) water deficit. Because of similar areas in both years, we refrained from depicting proportional areas as for the full comparison.**

**Results:**

**This observation was confirmed when considering only pixels with extreme (CWB anomaly < -2) or moderate (-2 < CWB < 0) water deficit (Figs. S11-S13) in 2003 and 2018. Although the differences of absolute areas decreased in this comparison, 2018 generally displayed larger areas with lowest quantiles compared to 2003. However, for EVI, subtle differences of opposite sign were observed for regions featuring extreme water deficit, while regions with moderate water deficit expressed similar patterns as for NDVI (Fig. S13 compared to Fig. S12).**

**Discussion, section 4.2.2**

**As a first step into this direction, we only compared VI-quantiles of regions that featured extreme or moderate water deficit in both years, thus only considering regions representative of the exactly same ecosystems. On average, we again observed a higher share of low quantiles in 2018 compared to 2003 (Figs. S12 and S13). Only for the EVI in regions featuring extreme drought, 2003 featured a slightly higher share of lower quantiles compared to 2018. Taken together, it**

*nevertheless seems that even when only considering the same regions for the comparison between both years, the impact of the 2018-drought supersedes the one of 2003. Interestingly, clear differences were only observed for forest ecosystems. This might indicate so-called drought legacy effects (see also section 4.2.4) as a consequence of preceding extreme droughts, such as the one of 2015 after which an increased forest mortality and growth decline was observed in southern Germany and other parts of Central Europe (Buras et al., 2018).*

*Conclusion:*

*Finally, in addition to quantifying impacts of the drought 2018 on European ecosystems our results possibly mirror forest drought legacies from preceding drought events. Moreover, additional legacy effects of forest ecosystems are likely to occur in course of the next years.*

Fig. 7: include statistics to indicate whether the difference between both years differed between the vegetation types.

*Reply: Since we only have one difference for each vegetation type, we struggle to provide a statistic that allows for testing whether the difference between the two years differ among the vegetation types. We have therefore just added a qualitative statement in the results section.*

**In comparison, the differences between 2003 and 2018 were highest for pastures, followed by arable land, broadleaved forests, mixed forests, and coniferous forests.**

Discussion:
l. 400: should this be "5 (±1) percent"?

Reply: This statement refers to the percentages of pixels that featured a positive CWB while the CWB anomaly was below 0 (5 percent) or below -2 (1 percent). These explanations are provided in the caption of Fig. S13 which is referred to in the same sentence. To clarify, we have added:

**for details see Fig. S13.**

l.407ff: it could also be mentioned in this paragraph that choice of VI can influence differences in responses between vegetation types (e.g. when some indices like NDVI detect drought better in grasslands than in forests, and vice versa for other VIs).

*Reply: We agree and have added the following sentence to this paragraph:*

**Consequently, the choice of VI may also affect the differences in the responses of different vegetation types.**

l.450 ff: choice of VI may also matter here (see previous remark). You find a similar response in EVI, supporting your statements. This is worth mentioning here.

*Reply: We have added the following statement to the corresponding paragraph:*

**As mentioned in section 4.2.1 the choice of VI may alter the differences in the responses of different vegetation types to drought. However, the fact that NDVI and EVI indicated differences of similar sign and magnitude among the considered ecosystems, supports our interpretation. Nevertheless, further remote sensing products such as the solar-induced fluorescence may provide additional information on ecosystem-specific drought responses.**